# Last-Iterate Convergence of Regularized Gradient Methods for Stochastic Monotone Variational Inequalities

**Shinji Ito** [1] [2]  **Taira Tsuchiya** [1] [2]  **Kaito Ariu** [3]  **Kenshi Abe** [3]

## Abstract

We study last-iterate convergence for stochastic smooth and monotone variational inequalities (VIs), a framework that captures convex-concave saddle points and Nash equilibrium computation in monotone games with noisy payoff feedback. In contrast to the well-understood average-iterate guarantees, anytime last-iterate guarantees in stochastic settings remain limited, despite their relevance for uncoupled learning dynamics that output a single current strategy. We analyze two single-call regularized methods, the *regularized gradient (RG)* and the *regularized optimistic gradient (ROG)* methods, and establish anytime last-iterate convergence rates in terms of the squared gap function. For monotone VIs, RG attains $O(t^{-2/5})$ while ROG achieves the variance-adaptive rate $O(\sigma^{4/5}t^{-2/5}+t^{-1})$, where $\sigma^2$ is the noise variance. For $\lambda$-strongly monotone VIs, ROG yields $O(\sigma^2/(\lambda^2 t)+t^{-c})$ for any constant $c \geq 2$. These results give anytime last-iterate guarantees without knowing the horizon and show that optimism improves convergence in the low-noise regime.

## 1. Introduction

Variational inequalities (VIs) provide a unified framework for a broad range of problems in optimization, game theory, and machine learning. Given a compact convex set $\mathcal{X} \subseteq \mathbb{R}^d$ and a continuous operator $V : \mathcal{X} \to \mathbb{R}^d$, the variational inequality problem seeks a point $x^* \in \mathcal{X}$ such that $\langle V(x^*), x - x^* \rangle \geq 0$ for all $x \in \mathcal{X}$. This formulation captures convex optimization (where $V = \nabla f$ for a convex function $f$), convex-concave saddle-point problems, and Nash equilibrium computation in multiplayer

games (Facchinei & Pang, 2003; Harker & Pang, 1990).

A particularly important application of VIs arises in *learning in games*, where multiple players repeatedly interact and update their strategies based on observed payoffs (Cesa-Bianchi & Lugosi, 2006). In this setting, the joint strategy profile $x = (x_1, \ldots, x_n)$ lies in the product space $\mathcal{X} = \mathcal{X}_1 \times \cdots \times \mathcal{X}_n$, and the operator $V$ is constructed from the individual payoff gradients. For monotone games (a broad class that includes convex-concave zero-sum games and concave potential games), finding a Nash equilibrium corresponds exactly to solving the associated VI. This connection has motivated extensive research on gradient-based methods that converge to equilibria, with applications ranging from generative adversarial networks (GANs) (Goodfellow et al., 2014; Gidel et al., 2019) to multi-agent reinforcement learning (Zhang et al., 2021).

In many modern applications, we only have access to noisy estimates of the operator $V$. In the context of learning in games, this noise arises naturally: players may only observe stochastic payoffs (e.g., from sampled opponents or random environments), estimate gradients from bandit feedback, or face uncertainty in their opponents' strategies. This motivates the study of *stochastic* variational inequalities, where at each iteration $t$, the algorithm observes a noisy feedback $\hat{g}_t$ satisfying $\mathbf{E}[\hat{g}_t | x_t] = V(x_t)$.

A fundamental question in this setting concerns the convergence of the *last iterate* $x_T$ rather than the average iterate $\bar{x}_T = \frac{1}{T} \sum_{t=1}^{T} x_t$. While average-iterate convergence is well-understood (Nemirovski, 2004; Juditsky et al., 2011), last-iterate convergence is often more desirable in practice: it avoids the need to store all past iterates, provides a single actionable solution, and is essential for applications where the algorithm's output must be used in real-time. In learning in games, last-iterate convergence is particularly important because players must commit to a specific strategy at each round; they cannot play a time-average of past strategies. Moreover, last-iterate convergence ensures that the actual behavior of players stabilizes near equilibrium, rather than merely averaging out over time.

Another important consideration in learning in games is whether the learning dynamics are *uncoupled*: each player's

---

[1]The University of Tokyo, Tokyo, Japan [2]RIKEN, Tokyo, Japan [3]CyberAgent, Tokyo, Japan. Correspondence to: Shinji Ito <shinji@mist.i.u-tokyo.ac.jp>.

*Proceedings of the 43rd International Conference on Machine Learning*, Seoul, South Korea. PMLR 306, 2026. Copyright 2026 by the author(s).

*Table 1.* Convergence rates for stochastic smooth monotone variational inequalities. "Hrz." denotes horizon (Any: anytime, Fix: fixed-horizon). $\lambda$-SM denotes $\lambda$-strong monotonicity. Our results are measured in terms of $\mathbf{E}[\mathrm{Gap}(x_t)^2]$; bounds on $\mathbf{E}[\mathrm{Gap}(x_t)]$ can be obtained via Jensen's inequality: $\mathbf{E}[\mathrm{Gap}(x_t)] \leq \sqrt{\mathbf{E}[\mathrm{Gap}(x_t)^2]}$. The step size $\eta_t$ and regularization weight $\gamma_t$ used for each setting are specified in the corresponding theorem. Constant factors and dependence on $D, L, U, G$ are omitted.

| Alg. | Hrz. | $\mathbf{E}[\mathrm{Gap}(x_t)^2]$ | Ref. |
|------|------|-----------------------------------|------|
| RG | Any | $t^{-\frac{2}{5}}$ | Thm. 5.1 |
| | Fix | $\left(\frac{\log T}{T}\right)^{\frac{1}{2}}$ | Thm. 5.4 |
| | Any | $\frac{1}{\lambda^2 t}$ | Thm. 5.6 ($\lambda$-SM) |
| ROG | Any | $\frac{\sigma^{\frac{4}{5}}}{t^{\frac{2}{5}}} + \frac{1}{t}$ | Thm. 6.1 |
| | Fix | $\sqrt{\frac{\sigma^2 \log T}{T} + \frac{(\log T)^2}{T^2}}$ | Thm. 6.3 |
| | Any | $\frac{1}{t^2}$ ($\sigma = 0$) | Cai & Zheng (2023)[1] |
| | Any | $\frac{c\sigma^2}{\lambda^2 t} + t^{-c}$ ($\forall c \geq 2$) | Thm. 6.5 ($\lambda$-SM) |

update rule should depend only on their own past actions and observed payoffs, without requiring knowledge of other players' strategies or payoff functions. Uncoupled dynamics are essential in practice because players typically cannot observe their opponents' private information. This rules out methods like the extragradient, which requires two gradient evaluations per iteration at different points; in a game setting, this would require players to coordinate on an intermediate strategy profile before taking the actual step. In contrast, single-call methods that use only one gradient evaluation per iteration naturally give rise to uncoupled dynamics when applied to games.

**Contributions.** In this paper, we analyze two regularized gradient methods for stochastic monotone VIs: the *regularized gradient (RG)* method and the *regularized optimistic gradient (ROG)* method. The RG method updates the iterate as

$$x_{t+1} = \Pi_{\mathcal{X}}\left((1 - \gamma_t)x_t - \eta_t \hat{g}_t\right), \qquad (1)$$

where $\Pi_{\mathcal{X}}$ denotes the Euclidean projection onto $\mathcal{X}$, $\gamma_t \geq 0$ is a regularization parameter, and $\eta_t > 0$ is the step size. The shrinkage term $(1 - \gamma_t)x_t$ pulls the iterate toward the origin, which acts as a time-varying regularization that stabilizes the dynamics. The ROG method combines this regularization with the optimistic gradient technique (Popov, 1980; Cai & Zheng, 2023), maintaining two sequences $\{x_t\}$ and $\{y_t\}$:

$$\begin{aligned} y_{t+1} &= \Pi_{\mathcal{X}}\left((1 - \gamma_t)y_t - \eta_t \hat{g}_t\right), \\ x_{t+1} &= \Pi_{\mathcal{X}}\left((1 - \gamma_{t+1})y_{t+1} - \eta_{t+1}\hat{g}_t\right). \end{aligned} \qquad (2)$$

The key idea of optimistic methods is to reuse the previous gradient $\hat{g}_t$ in the update of $x_{t+1}$, which allows the algorithm to exploit smoothness and cancel out errors when

consecutive gradients are similar. Both methods can be implemented as uncoupled learning dynamics in monotone games, since each player's update depends only on their own strategy and observed feedback.

Our main results are summarized in Table 1. The gap function $\mathrm{Gap}(\hat{x}) = \max_{x \in \mathcal{X}}\langle V(\hat{x}), \hat{x} - x\rangle$ measures the distance to a solution: $\mathrm{Gap}(\hat{x}) = 0$ if and only if $\hat{x}$ solves the VI. In the table, "Any" denotes anytime convergence, where the bound on $\mathbf{E}[\mathrm{Gap}(x_t)^2]$ holds for all $t \geq 1$ without prior knowledge of the total number of iterations. In contrast, "Fix" denotes fixed-horizon convergence, where the algorithm requires the horizon $T$ to be specified in advance and the bound holds only at the final iterate $x_T$.

We measure convergence in terms of the squared gap function $\mathbf{E}[\mathrm{Gap}(x_t)^2]$ rather than the gap function $\mathbf{E}[\mathrm{Gap}(x_t)]$. By Jensen's inequality, $\mathbf{E}[\mathrm{Gap}(x_t)] \leq \sqrt{\mathbf{E}[\mathrm{Gap}(x_t)^2]}$, so our bounds immediately imply bounds on $\mathbf{E}[\mathrm{Gap}(x_t)]$ by taking square roots. We focus on $\mathbf{E}[\mathrm{Gap}(x_t)^2]$ because there may be a fundamental gap between $\mathbf{E}[\mathrm{Gap}(x_t)^2]$ and $\mathbf{E}[\mathrm{Gap}(x_t)]^2$, as suggested by recent upper and lower bounds for uncoupled learning in matrix games (Fiegel et al., 2025). More precisely, Jensen's inequality gives $\mathbf{E}[\mathrm{Gap}(x_t)^2] \geq \mathbf{E}[\mathrm{Gap}(x_t)]^2$, so an upper bound on $\mathbf{E}[\mathrm{Gap}(x_t)^2]$ is strictly stronger, and thus harder to establish, than the corresponding bound on $\mathbf{E}[\mathrm{Gap}(x_t)]^2$. Bounding the squared gap moreover controls the second moment of $\mathrm{Gap}(x_t)$, which yields stronger concentration guarantees than a bound on the mean alone. Whether $\mathbf{E}[\mathrm{Gap}(x_t)]$ can converge faster than $\sqrt{\mathbf{E}[\mathrm{Gap}(x_t)^2]}$ remains an open question.

For the RG method applied to stochastic monotone VIs, we establish anytime last-iterate convergence at rate $O(t^{-2/5})$ in terms of the squared gap function $\mathbf{E}[\mathrm{Gap}(x_t)^2]$ (Theorem 5.1). When the time horizon $T$ is known in advance, the rate improves to $O(\sqrt{\log T / T})$ (Theorem 5.4). For $\lambda$-strongly monotone VIs, RG achieves $O(1/(\lambda^2 t))$ (Theorem 5.6).

The ROG method achieves *variance-adaptive* rates that interpolate between the noisy and noiseless regimes. In the anytime setting, ROG achieves $O(\sigma^{4/5} t^{-2/5} + t^{-1})$, where $\sigma^2$ is the noise variance (Theorem 6.1). When the noise is small ($\sigma \to 0$), this rate approaches $O(t^{-1})$. This variance-adaptive behavior is a significant advantage over the RG method, whose rate $O(t^{-2/5})$ does not improve even when the noise vanishes. For strongly monotone VIs, ROG achieves $O(\sigma^2/(\lambda^2 t) + t^{-c})$ for any constant $c > 1$ (Theorem 6.5), showing that the initial condition contribution can decay arbitrarily fast.

---

[1] While we present this method as a special case of the ROG framework in the noiseless setting ($\sigma = 0$), it can also be understood as an approximation of the Halpern iteration (Halpern, 1967).

Additionally, we provide regret bounds for the RG method in Section 7, and numerical experiments validating our theoretical results are presented in Appendix F.

A central message of our results is that *simple, single-call* regularized gradient methods already suffice for anytime last-iterate convergence. The most closely related prior guarantees, due to Abe et al. (2024) and Abe et al. (2025), rely on more elaborate anchoring-based designs with periodically reinitialized schedules tuned to a fixed horizon $T$, and achieve $\mathbf{E}[\mathrm{Gap}(x_T)] = \tilde{O}(T^{-1/10})$ and $\tilde{O}(T^{-1/7})$, respectively. By contrast, RG and ROG use a single gradient evaluation per iteration together with a simple jointly decreasing step size and regularization weight, yet attain the faster rate $\mathbf{E}[\mathrm{Gap}(x_t)] = O(t^{-1/5})$ in the stronger *anytime* setting, without any knowledge of the horizon. Two further results bear directly on our setting. Chen & Mazumdar (2024) obtain an $O(T^{-1/6})$ stochastic last-iterate rate for constrained monotone VIs in a fixed-horizon setting, using a generalized Frank–Wolfe method built on a linear-minimization oracle rather than the projected gradient updates we employ. Alacaoglu et al. (2024) obtain a stochastic $\widetilde{O}(T^{-1/4})$ rate for the resolvent/operator residual, equivalently $\widetilde{O}(\varepsilon^{-4})$ stochastic first-order oracle complexity, for the monotone/cohypomonotone setting. Their guarantee is stated for the operator residual rather than the gap function and is calibrated to a prescribed horizon or target accuracy, and therefore is not an anytime guarantee in the sense considered here. We discuss these and other comparisons in detail in Appendix A.

**Analysis overview.** Our analysis is based on tracking the distance between the iterate $x_t$ and the solution $x_t^*$ of a regularized VI. Specifically, we consider the regularized operator $V_t(x) = V(x) + \omega_t x$, where $\omega_t = \gamma_t/\eta_t$. Since $V_t$ is $\omega_t$-strongly monotone, there exists a unique solution $x_t^* \in \mathcal{X}$ satisfying $\langle V_t(x_t^*), x - x_t^* \rangle \geq 0$ for all $x \in \mathcal{X}$ (Harker & Pang, 1990; Facchinei & Pang, 2003). The key observation is that the RG update (1) can be viewed as performing projected gradient descent on the regularized VI, which converges to $x_t^*$. This perspective, reminiscent of the proximal point interpretation of optimistic methods (Mokhtari et al., 2020), allows us to establish contraction toward $x_t^*$.

The regularization strength $\omega_t$ controls a bias-variance trade-off: stronger regularization (larger $\omega_t$) makes the dynamics more stable and easier to analyze, but introduces a larger bias since $x_t^*$ deviates from the true solution $x^*$. Conversely, weaker regularization reduces bias but makes convergence harder to control. Our parameter choices balance this trade-off to achieve the stated convergence rates.

For the ROG method, the analysis is more intricate because the optimistic structure must be carefully combined with the regularization. The parameter design must additionally account for the noise variance $\sigma^2$ to achieve variance-adaptive rates. When $\sigma$ is small, we can use weaker regularization and larger step sizes, which leads to faster convergence.

In the strongly monotone case, the operator $V$ itself provides sufficient stability, so regularization is unnecessary ($\gamma_t = 0$). The unique solution $x^*$ is well-defined without regularization, and standard analysis techniques for strongly monotone VIs (Tseng, 1995; Gidel et al., 2019) can be adapted to our setting.

## 2. Notation and Problem Setup

We use $\|\cdot\|$ to denote the Euclidean norm. We write $a \lesssim b$ to mean $a \leq Cb$ for some absolute constant $C > 0$. Let $\mathcal{X} \subseteq \mathbb{R}^d$ be compact and convex, and define its diameter as $D = \max_{x,y \in \mathcal{X}} \|x - y\|$. We assume $0 \in \mathcal{X}$ without loss of generality. Let $V : \mathcal{X} \to \mathbb{R}^d$ be a monotone and $L$-Lipschitz continuous operator, i.e.,

$$\langle V(x) - V(y), x - y \rangle \geq 0 \qquad (3)$$

and $\|V(x) - V(y)\| \leq L\|x - y\|$ for any $x, y \in \mathcal{X}$. We refer to such a VI as $L$-smooth. Since $\mathcal{X}$ is compact and $V$ is continuous, the quantity $U := \max_{x \in \mathcal{X}} \|V(x)\|$ is well-defined and finite by the extreme value theorem; in particular, $\|V(x)\| \leq U$ for any $x \in \mathcal{X}$.

At round $t$, the decision is $x_t \in \mathcal{X}$ and the algorithm receives a noisy feedback $\hat{g}_t \in \mathbb{R}^d$ with $\mathbf{E}[\hat{g}_t|x_t] = g_t := V(x_t)$, $\mathbf{E}[\|\hat{g}_t\|^2|x_t] \leq G^2$, and $\mathbf{E}[\|\hat{g}_t - g_t\|^2|x_t] \leq \sigma^2$. We use the gap function $\mathrm{Gap}(\hat{x}) = \max_{x \in \mathcal{X}} \langle V(\hat{x}), \hat{x} - x \rangle \geq 0$ as the optimality measure. Note that the iterate $x_{t+1}$ is determined by the feedback observed up to round $t$. In the game interpretation, this means that after round $T$ each player can commit to the action rule $x_{T+1}$ using only information available up to time $T$. The goal is to find $x \in \mathcal{X}$ such that $\mathrm{Gap}(x)$ is small. More precisely, we focus on the (anytime) last-iterate convergence of $x_t$ with respect to $\mathbf{E}[\mathrm{Gap}(x_t)^2]$.

As discussed by Cai & Zheng (2023), this framework encompasses Nash equilibrium computation in monotone games as a special case. In this setting, $\mathcal{X}$ admits a decomposition into the product of strategy sets $\mathcal{X} = \mathcal{X}_1 \times \mathcal{X}_2 \times \cdots \times \mathcal{X}_n$. If each player $i$ has a differentiable utility function $u_i(x_i, x_{-i})$ that is concave in its own action, then the game induces the operator $V(x) = (-\nabla_{x_1} u_1(x), \ldots, -\nabla_{x_n} u_n(x))$, and the associated VI characterizes the first-order optimality (Nash) conditions for the game. Matrix games (two-player zero-sum games with finite actions) form a further special case, where $\mathcal{X}_1$ and $\mathcal{X}_2$ are simplices over the players' actions and the operator $V$ is induced by a bilinear payoff matrix. In this matrix-game setting, stochastic feedback naturally arises when players only observe realized payoffs from sampled actions rather than the expected payoff under the mixed strategies. For example, in each round, the players sample

pure actions according to their mixed strategies and observe a single noisy payoff realization (e.g., the matrix entry plus additive noise), which yields an unbiased but stochastic estimate of the game payoff and hence of $V(x_t)$.

In some sections of this paper, we consider the case where $V$ is *strongly monotone*. We say that $V : \mathcal{X} \to \mathbb{R}^d$ is $\lambda$-strongly monotone ($\lambda$-SM) for some $\lambda > 0$ if the following inequality holds for all $x, y \in \mathcal{X}$:

$$\langle V(x) - V(y), x - y \rangle \geq \lambda \|x - y\|^2. \quad (4)$$

Note that $\lambda \leq L$ holds automatically under our assumptions: combining $\lambda$-strong monotonicity with the Cauchy–Schwarz inequality and $L$-Lipschitz continuity gives, for any $x \neq y$, $\lambda \|x - y\|^2 \leq \langle V(x) - V(y), x - y \rangle \leq \|V(x) - V(y)\| \|x - y\| \leq L\|x - y\|^2$, so $\lambda \leq L$.

*Remark* 2.1. The assumption $0 \in \mathcal{X}$ is made without loss of generality. Indeed, for any nonempty $\mathcal{X}$ and $V : \mathcal{X} \to \mathbb{R}^d$, one may select an arbitrary point $x_0 \in \mathcal{X}$ and define $\mathcal{X}' = \{x - x_0 \mid x \in \mathcal{X}\}$ and $V' : \mathcal{X}' \to \mathbb{R}^d$ by $V'(x) = V(x + x_0)$. The resulting problem instance is equivalent to the original and satisfies $0 \in \mathcal{X}'$.

## 3. Algorithms

### 3.1. Stochastic Regularized Gradient (RG) Method

We present the *regularized gradient (RG)* method. Initialize $x_1 = 0$ and update the iterate according to

$$x_{t+1} = \Pi_{\mathcal{X}} \left( (1 - \gamma_t)x_t - \eta_t \hat{g}_t \right), \quad (5)$$

where $\Pi_{\mathcal{X}}$ denotes the Euclidean projection onto $\mathcal{X}$, and $\gamma_t \geq 0$ and $\eta_t > 0$ are algorithm parameters specified subsequently. In the context of monotone games where $\mathcal{X} = \mathcal{X}_1 \times \mathcal{X}_2 \times \cdots \times \mathcal{X}_n$, this algorithm admits a decentralized implementation (i.e., uncoupled learning dynamics), since each player $i \in [n]$ can perform the update using only its own strategy $x_{i,t}$ and feedback $\hat{g}_{i,t}$. The update may alternatively be interpreted as projected gradient descent on a time-varying regularized VI with operator $V_t(x) = V(x) + \omega_t x$ where $\omega_t = \gamma_t / \eta_t$, yielding a strongly monotone surrogate problem at each iteration (Harker & Pang, 1990; Facchinei & Pang, 2003).

### 3.2. Stochastic Regularized Optimistic Gradient (ROG) Method

We present the *regularized optimistic gradient (ROG)* method. Initialize $x_1 = y_1 = 0$ and update the iterates according to

$$y_{t+1} = \Pi_{\mathcal{X}} \left( (1 - \gamma_t)y_t - \eta_t \hat{g}_t \right), \quad (6)$$
$$x_{t+1} = \Pi_{\mathcal{X}} \left( (1 - \gamma_{t+1})y_{t+1} - \eta_{t+1}\hat{g}_t \right). \quad (7)$$

This single-call optimistic variant reuses the gradient $\hat{g}_t$ in the computation of $x_{t+1}$. The method coincides with the

accelerated optimistic gradient (AOG) algorithm of Cai & Zheng (2023) in the deterministic setting with $\eta_t = \Theta(1)$ and $\gamma_t = \Theta(1/t)$, and is closely related to classical optimistic gradient methods (Popov, 1980).

**Connection between RG and ROG.** Both methods are built on the same regularized operator framework of Section 4: they use the time-varying regularization $V_t(x) = V(x) + \omega_t x$ with $\omega_t = \gamma_t / \eta_t$, and their analyses rely on the same stability and gap-decomposition lemmas (Lemmas 4.1 and 4.2). The difference is that ROG adds optimistic gradient reuse on top of RG's regularization: in (7), the gradient $\hat{g}_t$ is reused in the update of $x_{t+1}$, so that the errors of consecutive gradient estimates partially cancel. This difference is reflected in the recursions. The RG recursion (Lemma 5.2) carries a noise term of order $G^2\eta_t^2$, where $G^2$ bounds $\mathbf{E}[\|\hat{g}_t\|^2]$ and thus retains a contribution from $\|V(x_t)\|^2$ even as $\sigma \to 0$. In contrast, the ROG recursion (Lemma 6.2) carries a noise term of order $\sigma^2\eta_t^2$, which depends only on the variance and vanishes in the noiseless limit. This structural difference is precisely what enables the variance-adaptive rate of ROG: when $\sigma$ is small, the noise term becomes negligible and ROG attains $O(t^{-1})$, whereas RG remains at $O(t^{-2/5})$ regardless of $\sigma$.

## 4. Analysis Framework

This section introduces the regularized operator framework and key lemmas that underpin the analysis of both the RG and ROG methods.

### 4.1. Regularized Operators

Throughout our analysis, we employ a regularization technique that transforms the original monotone VI into a sequence of strongly monotone subproblems. Specifically, we define the regularized operator

$$V_t(x) = V(x) + \omega_t x, \quad \text{where} \quad \omega_t = \frac{\gamma_t}{\eta_t}. \quad (8)$$

Since $V_t(x)$ is $\omega_t$-strongly monotone, there exists a unique solution $x_t^* \in \mathcal{X}$ to the variational inequality problem defined by $V_t(x)$, i.e., the point $x_t^*$ such that

$$\langle V_t(x_t^*), x_t^* - x \rangle = \langle V(x_t^*) + \omega_t x_t^*, x_t^* - x \rangle \leq 0 \quad (9)$$

for all $x \in \mathcal{X}$ (see, e.g., Corollary 3.2 of Harker & Pang, 1990). The regularized gap function is defined as

$$\text{Gap}_t(\hat{x}) = \max_{x \in \mathcal{X}} \langle V_t(\hat{x}), \hat{x} - x \rangle \geq 0. \quad (10)$$

The strong monotonicity of $V_t$ implies the following key inequality:

$$
\begin{aligned}
\mathrm{Gap}_t(\hat{x}) &\geq \langle V_t(\hat{x}), \hat{x} - x_t^* \rangle = \langle V(\hat{x}) + \omega_t \hat{x}, \hat{x} - x_t^* \rangle \\
&\geq \langle V(x_t^*) + \omega_t \hat{x}, \hat{x} - x_t^* \rangle \\
&\geq \langle -\omega_t x_t^* + \omega_t \hat{x}, \hat{x} - x_t^* \rangle = \omega_t \|\hat{x} - x_t^*\|^2,
\end{aligned}
\tag{11}
$$

where the second inequality follows from the monotonicity of $V$ and the last inequality follows from (9).

### 4.2. Key Lemmas

The following lemmas form the foundation of our convergence analysis. They establish how the regularized solutions $x_t^*$ evolve over time and how the original gap function relates to the distance from $x_t^*$.

**Lemma 4.1** (Stability of regularized solutions). *Assume that $\omega_t > 0$. Then, it holds that*

$$
\|x_{t+1}^* - x_t^*\| \leq \left|1 - \frac{\omega_{t+1}}{\omega_t}\right| D.
\tag{12}
$$

This lemma shows that as the regularization parameters stabilize (i.e., $\omega_{t+1}/\omega_t \to 1$), the regularized solutions $x_t^*$ converge. The proof is provided in Appendix C.

**Lemma 4.2** (Gap function bound). *For any $\hat{x} \in \mathcal{X}$ and for any $t$, we have*

$$
\mathrm{Gap}(\hat{x}) \leq (U + LD)\|\hat{x} - x_t^*\| + D^2 \omega_t.
\tag{13}
$$

This lemma decomposes the gap into two terms: (1) a distance term $(U+LD)\|\hat{x} - x_t^*\|$ measuring how far the iterate is from the regularized solution, and (2) a bias term $D^2 \omega_t$ reflecting the deviation of $x_t^*$ from the true solution due to regularization. As $\omega_t \to 0$, the bias vanishes and $x_t^*$ approaches the true solution. The proof is provided in Appendix B.

## 5. Analysis of the RG Method

### 5.1. Anytime Convergence

This section establishes the following convergence result:

**Theorem 5.1.** *Let $\{x_t\}$ be the sequence generated by (5) with $\gamma_t = t^{-4/5}$ and $\eta_t = \frac{D}{G} t^{-3/5}$. Then, it holds for any $t \geq 1$ that*

$$
\mathbf{E}\left[\mathrm{Gap}(x_t)^2\right] = O\left(c_1^2 t^{-2/5}\right),
\tag{14}
$$

*where $c_1 = D(U + G + LD)$.*

The following one-step recursion on $a_t = \mathbf{E}[\|x_{t-1}^* - x_t\|^2]$, the squared distance to the regularized solution, is the central building block of our analysis: combined with the stability bound on $\|x_t^* - x_{t-1}^*\|$ (Lemma 4.1) and a suitable parameter schedule, it yields the rate in Theorem 5.1.

**Lemma 5.2.** *Suppose that $x_t$ is generated by (5) with $\gamma_t > 0$. Then, $a_t \geq 0$ defined by $a_t = \mathbf{E}\left[\|x_{t-1}^* - x_t\|^2\right]$ satisfies*

$$
\begin{aligned}
a_{t+1} &\leq (1 - \gamma_t)a_t + \left(1 + \frac{1}{\gamma_t}\right)\|x_t^* - x_{t-1}^*\|^2 \\
&\quad + 2G^2\eta_t^2 + 2D^2\gamma_t^2.
\end{aligned}
$$

*Proof sketch.* Standard projected gradient descent analysis with comparator $x^* = x_t^*$ yields $\mathbf{E}[\|x_t^* - x_{t+1}\|^2 | x_t] \leq (1 - 2\gamma_t)\|x_t^* - x_t\|^2 + 2G^2\eta_t^2 + 2D^2\gamma_t^2$. To relate $\|x_t^* - x_t\|^2$ to $a_t$, we expand $\|x_t^* - x_t\|^2 = \|(x_t^* - x_{t-1}^*) + (x_{t-1}^* - x_t)\|^2 = \|x_t^* - x_{t-1}^*\|^2 + \|x_{t-1}^* - x_t\|^2 + 2\langle x_t^* - x_{t-1}^*, x_{t-1}^* - x_t \rangle$. The cross term is bounded by Cauchy–Schwarz and the AM-GM inequality: $2\langle x_t^* - x_{t-1}^*, x_{t-1}^* - x_t \rangle \leq 2\|x_t^* - x_{t-1}^*\|\|x_{t-1}^* - x_t\| \leq \gamma_t^{-1}\|x_t^* - x_{t-1}^*\|^2 + \gamma_t\|x_{t-1}^* - x_t\|^2$. Substituting back, the coefficient of $\|x_{t-1}^* - x_t\|^2$ becomes $(1 - 2\gamma_t)(1 + \gamma_t) \leq 1 - \gamma_t$. The complete proof is provided in Appendix C. $\square$

*Remark* 5.3 (Technical novelty). The tight handling of the cross term in the proof of Lemma 5.2 is the key technical improvement over prior analyses cast in the same regularized framework, notably Appendix C of Cai et al. (2023). Expanding $\|x_t^* - x_t\|^2$ as $\|x_t^* - x_{t-1}^*\|^2 + \|x_{t-1}^* - x_t\|^2$ plus a cross term, and controlling the cross term by the Cauchy–Schwarz and AM-GM inequalities tuned to $\gamma_t$, yields the tight contraction coefficient $(1 - \gamma_t)$ on $a_t$. The analogous step in prior work bounds the same quantity more loosely, of order $\|x_t^* - x_{t-1}^*\|$, which leads to a slower $O(t^{-1/3})$ rate under the corresponding parameter choices; our refinement improves this to $O(t^{-2/5})$. The technique is not specific to RG and is potentially applicable to other algorithms analyzed within this framework.

*Proof sketch of Theorem 5.1.* With the chosen parameters, Lemma 4.1 yields $\|x_t^* - x_{t+1}^*\| \leq D/(5t)$. Substituting into Lemma 5.2 gives $a_{t+1} \leq (1 - t^{-4/5})a_t + 5D^2 t^{-6/5}$. By induction, we establish $a_t \leq 10D^2 t^{-2/5}$ for all $t \geq 1$. Combining with Lemma 4.2 yields the stated bound. The complete proof is provided in Appendix C. $\square$

### 5.2. Fixed-Horizon Convergence

When the total number of iterations $T$ is known in advance, improved convergence rates can be achieved by using constant step sizes tuned to $T$.

**Theorem 5.4.** *For some $T \geq 2$, suppose that $\{x_t\}_{t=1}^T$ is generated by (5) with $\gamma_t = \gamma = \frac{\log T}{T}$ and $\eta_t = \eta = \frac{D\gamma^{3/4}}{\sqrt{MG}}$, where we denote $M = U + DL$. We then have*

$$
\mathbf{E}\left[\mathrm{Gap}(x_T)^2\right] = O\left(D^2 M\left(G\sqrt{\frac{\log T}{T}} + M\frac{\log T}{T}\right)\right).
$$

*Proof sketch.* With constant parameters $\gamma$ and $\eta$, the regularized solution $x_t^*$ remains fixed throughout, simplifying the recursion in Lemma 5.2 to $a_{t+1} \leq (1-\gamma)a_t + 2G^2\eta^2 + 2D^2\gamma^2$. Unrolling this recursion yields $a_T \lesssim (1-\gamma)^{T-1}D^2 + (G^2\eta^2 + D^2\gamma^2)/\gamma$. The choice $\gamma = (\log T)/T$ ensures $(1-\gamma)^{T-1} \lesssim 1/T$, while balancing the remaining terms through $\eta = D\gamma^{3/4}/\sqrt{MG}$ gives the stated rate. The complete proof is provided in Appendix C. $\qquad\square$

### 5.3. Strongly Monotone Case

We now consider the setting where $V$ is $\lambda$-strongly monotone. Under this assumption, there exists a unique solution $x^* \in \mathcal{X}$ to the variational inequality problem, characterized by $\mathrm{Gap}(x^*) = 0$ (see, e.g., Corollary 3.2 of Harker & Pang, 1990). Moreover, the following bound holds for all $\hat{x} \in \mathcal{X}$:

$$
\mathrm{Gap}(\hat{x}) \geq \langle V(\hat{x}), \hat{x} - x^* \rangle
$$
$$
\geq \langle V(x^*), \hat{x} - x^* \rangle + \lambda\|\hat{x} - x^*\|^2 \geq \lambda\|\hat{x} - x^*\|^2. \tag{15}
$$

**Lemma 5.5.** *Assume that $V$ is $\lambda$-strongly monotone for some $\lambda > 0$. Suppose that $x_t$ is generated by (5) with $\gamma_t = 0$. Then, $a_t \geq 0$ defined by $a_t = \mathbf{E}\left[\|x^* - x_t\|^2\right]$ satisfies*

$$
a_{t+1} \leq (1 - 2\eta_t\lambda)a_t + G^2\eta_t^2. \tag{16}
$$

The proof follows from the standard projected gradient descent analysis combined with strong monotonicity; see Appendix C.

**Theorem 5.6.** *Assume that $V$ is $\lambda$-strongly monotone for some $\lambda > 0$. Suppose that $\{x_t\}$ is generated by (5) with $\gamma_t = 0$ and $\eta_t = \frac{1}{\lambda t}$. We then have for all $t \geq 1$,*

$$
\mathbf{E}\left[\|x_t - x^*\|^2\right] = O\left(\frac{D^2\lambda^2 + G^2}{\lambda^2 t}\right),
$$

*and consequently $\mathbf{E}[\mathrm{Gap}(x_t)^2] = O((U + LD)^2(D^2\lambda^2 + G^2)/(\lambda^2 t))$.*

*Proof sketch.* Define $C = 2D^2 + \frac{4G^2}{\lambda^2}$. We establish the bound $a_t \leq \frac{C}{t}$ for $t \geq 2$ by induction on the recursion from Lemma 5.5. The base case $a_2 \leq D^2 \leq \frac{C}{2}$ holds by the definition of $C$. For the inductive step with $a_t \leq \frac{C}{t}$, applying the recursion with $\eta_t = (\lambda t)^{-1}$ yields

$$
a_{t+1} \leq \left(1 - \frac{2}{t}\right)\frac{C}{t} + \frac{G^2}{\lambda^2 t^2} \leq \frac{C}{t} - \frac{C}{t^2} \leq \frac{C}{t+1},
$$

where we used $G^2/\lambda^2 \leq C/4$ and $(t-1)(t+1) \leq t^2$. The final bound follows from (15), which gives $\mathrm{Gap}(\hat{x}) \leq (U + LD)\|\hat{x} - x^*\|$. The complete proof is provided in Appendix C. $\qquad\square$

## 6. Analysis of the ROG Method

### 6.1. Anytime Convergence

This section establishes the following convergence result:

**Theorem 6.1.** *Suppose that $\{x_t\}$ is given by (6) and (7) with $\gamma_t = \min\left\{\frac{1}{\sqrt{t}}, \left(\frac{DL}{\sigma t^2}\right)^{2/5}\right\}$ and $\eta_t = \frac{1}{6}\min\left\{\frac{1}{L}, \left(\frac{D^4}{L\sigma^4 t^3}\right)^{1/5}\right\}$. We then have*

$$
\mathbf{E}\left[\mathrm{Gap}(x_t)^2\right]
$$
$$
= O\left(D^2(U + LD)^2 \max\left\{t^{-1}, \left(\frac{\sigma}{LD}\right)^{4/5} t^{-2/5}\right\}\right) \tag{17}
$$

*for all $t$. Equivalently, we have $\mathbf{E}\left[\mathrm{Gap}(x_t)^2\right] = O\left(D^2(U + LD)^2/t\right)$ for $t \leq t_0 := (LD/\sigma)^{4/3}$ and $\mathbf{E}\left[\mathrm{Gap}(x_t)^2\right] = O\left(D^{6/5}(U + LD)^2\sigma^{4/5}L^{-4/5}t^{-2/5}\right)$ for $t \geq t_0$.*

The two cases in Theorem 6.1 reflect a *two-phase behavior* governed by the noise level, with transition at $t_0 = (LD/\sigma)^{4/3}$. In the early phase $t \leq t_0$, the step size is held *constant* at $\eta_t = 1/(6L)$ while the regularization decays rapidly as $\gamma_t = t^{-1/2}$. It is this aggressive decay of the regularization, rather than a decaying step size, that drives the fast $O(t^{-1})$ rate, recovering the noiseless rate of Cai & Zheng (2023) on this phase. Once the accumulated noise becomes dominant, the algorithm enters the late phase $t \geq t_0$, where both parameters decay ($\eta_t \propto t^{-3/5}$ and $\gamma_t \propto t^{-4/5}$) and the variance-limited rate $O(\sigma^{4/5}t^{-2/5})$ takes over. Since the transition point scales as $t_0 \propto \sigma^{-4/3}$, the favorable $O(t^{-1})$ phase lasts longer as the noise shrinks, and extends to all $t$ in the noiseless limit $\sigma \to 0$.

The following recursion bound is central to our analysis.

**Lemma 6.2.** *Define $a_t = \mathbf{E}\left[\|y_t - x_{t-1}^*\|^2 + \frac{1}{2}\|x_{t-1} - y_t\|^2\right]$. Assume that $0 < \gamma_t \leq \frac{1}{2}$ and $0 < \eta_t \leq \frac{1}{6L}$. We then have*

$$
a_{t+1} \leq \left(1 - \frac{\gamma_t}{2}\right)a_t + \left(1 + \frac{2}{\gamma_t}\right)\|x_t^* - x_{t-1}^*\|^2
$$
$$
+ 12\sigma^2\eta_t^2. \tag{18}
$$

The choice of potential $a_t$ here departs from the RG analysis, where tracking $\mathbf{E}[\|x_t - x_t^*\|^2]$ alone suffices (Lemma 5.2). For ROG, the two-sequence structure, in which $x_t$ and $y_t$ are updated with the reused gradient $\hat{g}_t$, does not yield a closed recursion in $\mathbf{E}[\|x_t - x_t^*\|^2]$. We therefore use the composite potential $a_t = \mathbf{E}[\|y_t - x_{t-1}^*\|^2 + \frac{1}{2}\|x_{t-1} - y_t\|^2]$, which couples the two sequences: the first term tracks the progress of $y_t$ toward the regularized solution, while the second penalizes the discrepancy $\|x_{t-1} - y_t\|$ that the optimistic step exploits. This composite form is what makes

the recursion compatible with the optimistic structure under time-varying regularization, and, as a by-product, controls both $\|y_t - x_t^*\|^2$ and $\|x_t - y_t\|^2$, so that the convergence guarantees extend to the auxiliary sequence $\{y_t\}$ as well.

*Proof sketch.* Let $\omega_t = \gamma_t/\eta_t$. We may write (6) as $y_{t+1} = \Pi_{\mathcal{X}}(y_t - \eta_t(\hat{g}_t + \omega_t y_t))$ and (7) as $x_t = \Pi_{\mathcal{X}}(y_t - \eta_t(\hat{g}_{t-1} + \omega_t y_t))$. Applying the standard projected gradient descent inequality to these two updates, with comparators $x_t^*$ and $y_{t+1}$ respectively, gives

$$2\eta_t \langle \hat{g}_t + \omega_t y_t, y_{t+1} - x_t^* \rangle$$
$$\leq \|y_t - x_t^*\|^2 - \|y_{t+1} - x_t^*\|^2 - \|y_t - y_{t+1}\|^2, \quad (19)$$
$$2\eta_t \langle \hat{g}_{t-1} + \omega_t y_t, x_t - y_{t+1} \rangle$$
$$\leq \|y_t - y_{t+1}\|^2 - \|x_t - y_{t+1}\|^2 - \|y_t - x_t\|^2. \quad (20)$$

Adding (19) and (20) and rearranging isolates a *gradient-difference term* $\langle \hat{g}_t - \hat{g}_{t-1}, x_t - y_{t+1} \rangle$ (arising from the optimistic reuse of $\hat{g}_t$) and a *contraction term* $\langle \hat{g}_t + \omega_t y_t, x_t^* - x_t \rangle$:

$$\|y_{t+1} - x_t^*\|^2 + \tfrac{1}{2}\|x_t - y_{t+1}\|^2$$
$$\leq 2\eta_t \langle \hat{g}_t - \hat{g}_{t-1}, x_t - y_{t+1} \rangle$$
$$+ 2\eta_t \langle \hat{g}_t + \omega_t y_t, x_t^* - x_t \rangle + \|y_t - x_t^*\|^2. \quad (21)$$

The gradient-difference term is bounded via Young's inequality, $2\eta_t \langle \hat{g}_t - \hat{g}_{t-1}, x_t - y_{t+1} \rangle \leq 2\eta_t^2 \|\hat{g}_t - \hat{g}_{t-1}\|^2 + \tfrac{1}{2}\|x_t - y_{t+1}\|^2$, combined with the smoothness of $V$ and the noise bound:

$$\mathbf{E}[\|\hat{g}_{t-1} - \hat{g}_t\|^2]$$
$$\leq 6\sigma^2 + 6L^2 \, \mathbf{E}[\|x_{t-1} - y_t\|^2 + \|x_t - y_t\|^2]. \quad (22)$$

This is the origin of the $12\sigma^2\eta_t^2$ noise term appearing in the recursion. The contraction term is handled by the monotonicity of $V$ together with the optimality (9) of $x_t^*$, which yields

$$2 \, \mathbf{E}[\langle \hat{g}_t + \omega_t y_t, x_t^* - x_t \rangle \mid x_t, y_t]$$
$$\leq \omega_t (\|x_t - y_t\|^2 - \|x_t^* - y_t\|^2). \quad (23)$$

Since $\gamma_t = \omega_t \eta_t$, the resulting term $-\gamma_t \|x_t^* - y_t\|^2$ combines with $\|y_t - x_t^*\|^2$ to produce the contraction factor $(1 - \gamma_t)$ on $\|y_t - x_t^*\|^2$. Finally, using $\eta_t \leq 1/(6L)$ to absorb the $\|x_t - y_t\|^2$ terms and Lemma B.2 to pass from $x_t^*$ to $x_{t-1}^*$ (which produces the $(1 + 2/\gamma_t)\|x_t^* - x_{t-1}^*\|^2$ term) gives the stated recursion. The complete proof is provided in Appendix D. $\square$

*Proof sketch of Theorem 6.1.* Denote $M = U + LD$ and $t_0 = (DL/\sigma)^{4/3}$. From Lemma 4.2, we have $\mathbf{E}[\mathrm{Gap}(x_t)^2] \lesssim M^2 a_{t+1} + D^4 \omega_t^2$. For $t \leq t_0$, use $\gamma_t = t^{-1/2}$, $\eta_t = (6L)^{-1}$. For $t > t_0$, use $\gamma_t = t_0^{3/10} t^{-4/5}$, $\eta_t = (D^4/(L\sigma^4 t^3))^{1/5}/6$.

Combining Lemmas 6.2 and 4.1 with bounds on $\|x_t^* - x_{t-1}^*\|$ from Lemma B.1, the recursion simplifies to $a_{t+1} \leq (1 - 1/(2\sqrt{t}))a_t + 3D^2 t^{-3/2}$ for $t \leq t_0$, yielding $a_t \leq 24D^2/t$ by induction. For $t > t_0$, the recursion becomes $a_{t+1} \leq (1 - t_0^{3/10}/(2t^{4/5}))a_t + 2D^2 t_0^{-3/10} t^{-6/5}$, yielding $a_t \leq 24D^2/(t_0^{3/5} t^{2/5})$ by induction. Combining with the gap bound gives the result. The complete proof is provided in Appendix D.2. $\square$

## 6.2. Fixed-Horizon Convergence

When the total number of iterations $T$ is known in advance, improved convergence rates can be achieved by using constant step sizes tuned to $T$.

**Theorem 6.3.** *For some $T \geq 30$, suppose that $\{x_t\}_{t=1}^T$ is generated by (6)–(7) with $\gamma_t = \gamma = \frac{4\log T}{T}$ and $\eta_t = \eta = \min\left\{ \frac{1}{6L}, \frac{D\gamma^{3/4}}{\sqrt{M\sigma}} \right\}$, where $M = U + DL$. We then have*

$$\mathbf{E}[\mathrm{Gap}(x_T)^2]$$
$$= O\left( D^2 M \sigma \sqrt{\frac{\log T}{T}} + \frac{D^2(M\log T)^2}{T^2} \right). \quad (24)$$

*Proof sketch.* With constant parameters $\gamma$ and $\eta$, the regularized solution $x_t^*$ remains fixed throughout, which simplifies the recursion in Lemma 6.2 to $a_{t+1} \leq (1 - \gamma/2)a_t + 12\sigma^2\eta^2$. Unrolling this recursion yields $a_T \lesssim (1 - 2(\log T)/T)^{T-1} D^2 + \sigma^2 \eta^2/\gamma \lesssim D^2/T^2 + \sigma^2\eta^2/\gamma$. The choice $\gamma = 4(\log T)/T$ ensures fast decay of the initial condition, while the variance-dependent choice of $\eta$ balances the remaining terms. The complete proof is provided in Appendix D. $\square$

## 6.3. Strongly Monotone Case

In this section, we consider the case where $V(x)$ is $\lambda$-strongly monotone. Then, there exists a unique solution $x^* \in \mathcal{X}$ to the variational inequality problem defined by $V(x)$ and $\mathcal{X}$. We also note that (15) holds as well. As discussed in the introduction, strongly monotone $V$ provides sufficient stability, so we can set $\gamma_t = 0$. In this case, the ROG algorithm (6)–(7) simplifies to

$$y_{t+1} = \Pi_{\mathcal{X}}(y_t - \eta_t \hat{g}_t), \quad (25)$$
$$x_{t+1} = \Pi_{\mathcal{X}}(y_{t+1} - \eta_{t+1} \hat{g}_t). \quad (26)$$

**Lemma 6.4.** *Assume that $V$ is $\lambda$-strongly monotone for some $\lambda > 0$. Define $a_t = \mathbf{E}[\|y_t - x^*\|^2 + \tfrac{1}{2}\|x_{t-1} - y_t\|^2]$. Assume that $0 < \eta_t \leq \frac{1}{6L}$. We then have*

$$a_{t+1} \leq (1 - \eta_t\lambda) a_t + 12\sigma^2\eta_t^2. \quad (27)$$

The proof follows a similar structure to Lemma 6.2, using strong monotonicity instead of the regularization term; see Appendix D.

**Theorem 6.5.** *Assume that $V$ is $\lambda$-strongly monotone for some $\lambda > 0$. Let $c \geq 2$ be an arbitrary constant and let $t_0 = \lceil 6cL/\lambda \rceil$. Suppose that $\{x_t\}$ is generated by (25)–(26) with $\eta_t = \frac{c}{\lambda(t+t_0)}$. We then have for all $t \geq 1$,*

$$\mathbf{E}\left[\|x_t - x^*\|^2\right] = O\left(\frac{c\sigma^2}{\lambda^2(t_0 + t)} + \frac{D^2}{(1 + t/t_0)^c}\right),$$

*and consequently $\mathbf{E}[\mathrm{Gap}(x_t)^2] = O((U + LD)^2(c\sigma^2/(\lambda^2(t_0 + t)) + D^2/(1 + t/t_0)^c))$.*

*Proof sketch.* Observe that $\eta_t = c/(\lambda(t + t_0)) \leq c/(\lambda t_0) \leq 1/(6L)$ for all $t \geq 1$. From Lemma 6.4, $a_{t+1} \leq (1 - c/(t + t_0))a_t + 12c^2\sigma^2/(\lambda^2(t + t_0)^2)$. Substituting $s = t + t_0$ and $b_s = a_t$, we obtain $b_{s+1} \leq (1 - c/s)b_s + 12c^2\sigma^2/(\lambda^2 s^2)$. An inductive argument with ansatz $b_s \leq A/s + B/s^c$, where $A = 12c^2\sigma^2/(\lambda^2(c - 1))$ and $B = D^2(t_0 + 1)^c$, combined with Lemma B.1, yields $a_t = O(c\sigma^2/(\lambda^2(t_0 + t)) + D^2/(1 + t/t_0)^c)$. The result follows from the bound $\mathrm{Gap}(x_t) \leq (U + LD)\|x_t - x^*\|$. The complete proof is provided in Appendix D. $\square$

*Remark* 6.6 (On exponential rates in the noiseless case). In the noiseless case $\sigma = 0$, one might expect the initial-condition term to decay exponentially rather than polynomially. Theorem 6.5 instead yields the polynomial term $t^{-c}$ with an arbitrarily large constant $c \geq 2$. The reason is that our parameter design prioritizes *anytime* guarantees that hold uniformly for all $\sigma \geq 0$, which constrains the step-size schedule. Obtaining exponential decay would require switching between a low-variance schedule, targeting exponential decay, and a high-variance schedule, but reconciling the two analyses at the transition point is delicate. Under a single unified schedule, the polynomial rate $t^{-c}$ with arbitrarily large $c$ is the best we obtain, and closing this gap is an interesting open problem.

*Remark* 6.7 (Convergence of auxiliary sequence). All convergence guarantees stated for $\{x_t\}$ in this section also hold for the auxiliary sequence $\{y_t\}$. This follows from the definition of the potential $a_t$, which controls both $\|y_t - x_t^*\|^2$ and $\|x_t - y_t\|^2$.

# 7. Regret Analysis

In this section, we complement the last-iterate convergence results with regret bounds, which are relevant for the online learning interpretation of our algorithms.

One motivation for studying regret here is the question, raised by Cai & Zheng (2023) in the noiseless setting, of whether no-regret learning and last-iterate convergence can be achieved *simultaneously* by a single algorithm. This section offers a partial answer in the stochastic setting: the RG method attains both sublinear regret ($O(T^{4/5})$, Proposition 7.1) and anytime last-iterate convergence (Theorem 5.1).

The regret rate is worse than the standard $O(\sqrt{T})$, which indicates that reconciling these two objectives under stochastic gradient feedback leaves room for improvement and is of independent interest.

## 7.1. Regret Definition

For a sequence of iterates $\{x_t\}_{t=1}^T$ and any comparator $u \in \mathcal{X}$, the *regret* is defined as

$$\mathrm{Reg}_T(u) = \sum_{t=1}^T \langle V(x_t), x_t - u \rangle. \tag{28}$$

In the game-theoretic setting where $V$ is the concatenation of individual gradients, $\mathrm{Reg}_T(u)$ corresponds to the cumulative loss incurred by the player relative to always playing the fixed action $u$. A sublinear regret bound $\mathrm{Reg}_T(u) = o(T)$ implies that the average payoff converges to that of the best fixed action in hindsight.

## 7.2. Regret Bound for the RG Method

The RG update (5) can be interpreted as projected gradient descent with the modified gradient $\hat{g}_t + \omega_t x_t$, where $\omega_t = \gamma_t/\eta_t$. Standard online gradient descent analysis yields the following result.

**Proposition 7.1.** *Let $\{x_t\}$ be the sequence generated by (5) with non-increasing $\{\eta_t\}$. For any $u \in \mathcal{X}$,*

$$\mathbf{E}[\mathrm{Reg}_T(u)] \leq \frac{D^2}{2\eta_T} + \sum_{t=1}^T \eta_t G^2 + \sum_{t=1}^T \omega_t D^2. \tag{29}$$

The proof follows from standard online gradient descent analysis; see Appendix E.

**Regret with anytime parameters.** For the parameter choice $\gamma_t = t^{-4/5}$ and $\eta_t = (D/G)t^{-3/5}$ from Theorem 5.1, we have $\omega_t = (G/D)t^{-1/5}$. Since $\sum_{t=1}^T t^{-1/5} = O(T^{4/5})$ and $D^2/(2\eta_T) = O(DGT^{3/5})$, Proposition 7.1 yields

$$\mathbf{E}[\mathrm{Reg}_T(u)] = O(DG \cdot T^{4/5}).$$

Due to regularization, the regret worsens from the $O(\sqrt{T})$ rate of standard projected gradient descent to $O(T^{4/5})$, which may be the price paid for achieving anytime last-iterate convergence.

*Remark* 7.2. In a game setting where $\mathcal{X} = \mathcal{X}_1 \times \cdots \times \mathcal{X}_n$, the regret bounds hold *regardless of how other players behave*: even if opponents play adversarially, a player using RG achieves sublinear regret, implying that their time-averaged payoff converges to at least that of the best fixed action. However, last-iterate convergence to Nash equilibrium requires coordinated dynamics: the results in Sections 5 and 6 assume that all players use the same algorithm.

## 8. Discussion and Future Work

We conclude by discussing several limitations of the present work and directions for future research.

**Lower bounds.** The convergence rates established in this paper are derived through upper bound analysis. To assess whether these rates are optimal, corresponding lower bounds are required. While Fiegel et al. (2025) established lower bounds for the bandit feedback setting, extending their techniques to stochastic gradient feedback appears challenging due to the increased information content of gradient observations (see Remark G.1 in the appendix for details). Establishing matching lower bounds for last-iterate convergence in stochastic monotone variational inequalities remains an open problem and would provide valuable insight into the fundamental limits of first-order methods in this setting.

**Parameter-free algorithms.** The step-size schedules proposed in this work require knowledge of problem parameters such as the Lipschitz constant $L$, the diameter $D$, and the noise level $\sigma$. In particular, achieving variance-adaptive convergence rates necessitates prior knowledge of the variance bound $\sigma^2$. Relaxing this requirement is desirable for practical applications. Adaptive methods inspired by Ada-Grad (Duchi et al., 2011) may offer a promising direction; however, it remains unclear whether such approaches can simultaneously tune both the regularization parameter $\gamma_t$ and the step size $\eta_t$ while preserving the theoretical guarantees.

**Beyond Euclidean geometry.** Our analysis is conducted entirely in the Euclidean norm setting. For certain problem structures, alternative geometries may be more natural. For instance, in matrix games where the feasible region consists of probability simplices, the $\ell_1$ norm on the primal space and the $\ell_\infty$ norm on the dual space provide a more appropriate geometric structure. In such settings, methods based on exponential weights or entropic regularization (Freund & Schapire, 1999; Cesa-Bianchi & Lugosi, 2006) are often preferable to projected gradient descent. More generally, extending the regularized gradient framework to mirror descent (Nemirovski & Yudin, 1983; Beck & Teboulle, 2003) would broaden the applicability of our results to a wider class of problems with non-Euclidean structure.

## Impact Statement

This paper presents theoretical advances in the analysis of optimization algorithms for variational inequalities. The primary contribution is methodological, establishing last-iterate convergence guarantees that improve our understanding of algorithm behavior. Variational inequalities have applications in game theory, multi-agent systems, and machine learning (e.g., training generative adversarial networks).

While this work is foundational in nature, we do not foresee direct negative societal consequences from our theoretical results.

## Acknowledgements

SI is supported by JSPS KAKENHI Grant Number JP25K03184 and by JST PRESTO, Japan, Grant Number JPMJPR2511. TT is supported by JSPS KAKENHI Grant Number JP24K23852 and partially supported by JSPS KAKENHI Grant Number JP26K21297. Kaito Ariu is supported by JSPS KAKENHI Grant Number JP25K21291.

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

# A. Related Work

Unless stated otherwise, the convergence rates presented here are with respect to $\text{Gap}(x_t)$.

**Last-iterate convergence for deterministic VIs.** The study of last-iterate convergence has received significant attention in recent years. For deterministic VIs, the extragradient method (Korpelevich, 1976) and optimistic gradient descent (Popov, 1980) are known to achieve $O(1/t)$ last-iterate convergence for monotone operators and linear convergence for strongly monotone operators (Tseng, 1995; Gidel et al., 2019; Mokhtari et al., 2020). Golowich et al. (2020) established $O(1/\sqrt{t})$ last-iterate convergence for the extragradient method in the monotone case, which was later improved to $O(1/t)$ by Gorbunov et al. (2022b) and Cai et al. (2022b). For accelerated rates, Yoon & Ryu (2021) and Cai & Zheng (2023) showed that optimistic methods can achieve $O(1/t^2)$ last-iterate convergence for smooth monotone VIs in the deterministic setting.

**Stochastic VIs.** In the stochastic setting, the picture is less complete. Early works such as (Koshal et al., 2010; 2013; Tatarenko & Kamgarpour, 2019) focused on the asymptotic convergence of iterative Tikhonov-regularization methods under noisy or payoff-based feedback. This line of work traces back to the stochastic approximation approach of Jiang & Xu (2008), and includes regularized smoothing schemes that drive the regularization to zero to select the least-norm solution (Yousefian et al., 2013), as well as smoothed stochastic extragradient methods with averaged-iterate rates (Yousefian et al., 2014). More recently, Azizian et al. (2021) analyzed optimistic mirror descent and established last-iterate convergence rates under a strong variational stability assumption. Hsieh et al. (2019) studied last-iterate convergence for stochastic extragradient methods, and Gorbunov et al. (2022a) provided convergence rates under various noise assumptions. Closest to our setting, Chen & Mazumdar (2024) analyzed a generalized Frank–Wolfe method for constrained monotone VIs, establishing an $O(T^{-1/2})$ last-iterate rate in the deterministic case and a slower $O(T^{-1/6})$ rate for a stochastic variant, both in a fixed-horizon setting; their mechanism builds on a linear-minimization oracle and a connection to smoothed fictitious play, in contrast to our regularized gradient updates. For nonmonotone problems, Pethick et al. (2023a) solve stochastic weak Minty VIs without increasing the batch size via a bias-corrected extragradient scheme, while Alacaoglu et al. (2024) analyze inexact Halpern iterations. As explained in Remark 4.3 of Alacaoglu et al. (2024), the stochastic oracle model employed by Pethick et al. (2023a) differs from the one considered in this paper. In the stochastic setting under cohypomonotonicity, Alacaoglu et al. (2024) reach an $\varepsilon$-accurate operator residual with $O(\varepsilon^{-4})$ stochastic first-order oracle calls, corresponding to an $O(T^{-1/4})$ rate. Several aspects distinguish their guarantee from ours: it is measured by the operator residual rather than the gap function; it is horizon-fixed, with the schedule tuned to a target accuracy. Moreover, their method couples an outer Halpern iteration with an inner subroutine that approximately solves a subproblem; while the stated bound concerns the outer scheme, anytime convergence of the inner subroutine is not established, so the overall method is not anytime in the strict sense provided by our analysis. In the *unconstrained* setting (i.e., $\mathcal{X} = \mathbb{R}^d$), where the operator residual $\|V(x)\|$ provides a natural convergence measure, Chen & Luo (2024) obtain near-optimal stochastic oracle complexity for stochastic minimax problems via anchored iteration, and Cai et al. (2022a) attain $O(\varepsilon^{-3})$ oracle complexity for cocoercive and Lipschitz-monotone inclusions via variance-reduced stochastic Halpern iterations. The constrained and unconstrained settings appear to differ in difficulty, with the constrained case being generally harder. Last-iterate guarantees for comonotone and cohypomonotone problems were also obtained by Pethick et al. (2023b) through anchoring and linear-interpolation (lookahead) techniques. However, most existing results either required knowledge of the time horizon $T$ to set algorithm parameters (fixed-horizon setting) or provided slower rates in the anytime setting.

**Learning in games.** In the context of learning in games, convergence to Nash equilibrium has been extensively studied from both the optimization and online learning perspectives (Cesa-Bianchi & Lugosi, 2006). No-regret learning algorithms such as multiplicative weights and follow-the-regularized-leader are known to converge to coarse correlated equilibria in general games, and to Nash equilibria in zero-sum games when measured by the average iterate (Freund & Schapire, 1999). However, last-iterate convergence is more challenging. Daskalakis & Panageas (2019) established asymptotic last-iterate convergence for optimistic gradient methods in constrained min-max optimization, and Wei et al. (2021) proved linear last-iterate convergence for optimistic multiplicative weights update (OMWU) and optimistic gradient descent ascent (OGDA) in bilinear games with unique equilibria. For strongly monotone and exp-concave games, Jordan et al. (2024) obtained doubly optimal, parameter-free no-regret learning rates under gradient feedback. In the more restrictive payoff-based (bandit) feedback model, Elvidge et al. (2026) combined optimistic gradient updates with simultaneous perturbation and a two-timescale design to obtain last-iterate convergence in monotone games with unbounded action spaces.

A related strand of work (Cen et al., 2023; 2024; Liu et al., 2023) achieved fast last-iterate convergence in structured two-player zero-sum settings by combining first-order policy updates with regularization and a shrinking regularization

weight. While these works provided last-iterate convergence guarantees in their respective models (e.g., normal-form / Markov / extensive-form games) in noiseless or full-information settings, our results are stated at the VI level and provide last-iterate guarantees under general stochastic gradient feedback.

**Matrix games under bandit feedback.** For matrix games (two-player zero-sum games with finite actions) under bandit feedback, where players only observe their realized payoffs rather than the full payoff vector, the problem becomes significantly harder due to the need to balance exploration and exploitation. While matrix games are a special case of monotone VIs in terms of problem structure, bandit feedback is more restrictive than the stochastic feedback model considered in this paper. O'Donoghue et al. (2021) initiated the study of matrix games with unknown payoff matrices under bandit feedback, showing that standard adversarial bandit algorithms fail to exploit the game structure. Cai et al. (2023) achieved the first anytime last-iterate convergence rate $\mathbf{E}[\text{Gap}(x_t)] = \tilde{O}(t^{-1/6})$ for matrix games with bandit feedback, which was later improved to $\tilde{O}(t^{-1/5})$ by Cai et al. (2025) using a reduction from average-iterate to last-iterate convergence. Fiegel et al. (2025) achieved $\mathbf{E}[\text{Gap}(x_T)^2] = O(T^{-1/2})$ last-iterate convergence for uncoupled learning in the fixed-horizon setting under bandit feedback, and also showed anytime $\mathbf{E}[\text{Gap}(x_t)^2] = O(t^{-1/2})$ convergence when players share common random bits (which is not strictly uncoupled). They further established a lower bound showing that the $O(t^{-1/4})$ rate cannot be improved in a setting closely related to anytime uncoupled learning, demonstrating a fundamental gap compared to the $O(T^{-1/2})$ rate achievable for average iterates. Note that these prior works on matrix games measure convergence in terms of $\mathbf{E}[\text{Gap}(x_t)]$ or $\text{Gap}(x_t)$, whereas our results (and those of Fiegel et al. (2025) for their lower bound) are stated for $\mathbf{E}[\text{Gap}(x_t)^2]$.

**Comparison with single-call dynamics for monotone games.** In the context of learning in monotone games under stochastic gradient feedback, Abe et al. (2024) and Abe et al. (2025) propose single-call first-order dynamics in a *fixed-horizon* regime: the step-size schedule is chosen as a (decreasing) function of the horizon $T$, and the analysis targets the final iterate $x_T$. These approaches yield last-iterate rates $\mathbf{E}[\text{Gap}(x_T)] = \tilde{O}(T^{-1/10})$ (Abe et al., 2024) and $\mathbf{E}[\text{Gap}(x_T)] = \tilde{O}(T^{-1/7})$ (Abe et al., 2025). Their design uses an anchoring strategy with periodic anchor updates, where the step-size schedule is re-initialized after each update; the perturbation magnitude is controlled via the distance to the current anchor, and the update timing is tuned as a function of $T$. In contrast, our analysis provides faster, *anytime* last-iterate bounds by jointly decreasing the step size and the regularization weight.

## B. Auxiliary Lemmas

**Lemma B.1.** *For any $\alpha > 0$ and $t > 0$, we have*

$$1 - \frac{\alpha}{t} \le \left(\frac{t}{t+1}\right)^{\alpha}. \tag{30}$$

*Proof.* If $t \le \alpha$, then the left-hand side is at most $0$ while the right-hand side is positive, so the inequality holds trivially. For $t > \alpha$, we have $\alpha/t \in (0, 1)$. Taking logarithms, the inequality is equivalent to $\ln(1 - \alpha/t) \le -\alpha \ln(1 + 1/t)$. From $\ln(1 - x) \le -x$ for $x \in (0, 1)$, we have

$$\ln\left(1 - \frac{\alpha}{t}\right) \le -\frac{\alpha}{t}. \tag{31}$$

From $\ln(1 + y) \le y$ for $y > 0$, we have

$$-\alpha \ln\left(1 + \frac{1}{t}\right) \ge -\frac{\alpha}{t}. \tag{32}$$

Combining these two inequalities, we obtain the desired result. □

**Lemma B.2.** *For any $x, y \in \mathbb{R}^d$ and $\varepsilon > 0$, the following holds:*

$$\|x + y\|^2 \le (1 + \varepsilon)\|x\|^2 + \left(1 + \frac{1}{\varepsilon}\right)\|y\|^2. \tag{33}$$

*Proof.* We have

$$\|x + y\|^2 = \|x\|^2 + \|y\|^2 + 2 \langle x, y \rangle \tag{34}$$

$$\leq \|x\|^2 + \|y\|^2 + 2\|x\| \cdot \|y\| \tag{Cauchy-Schwarz}$$

$$\leq \|x\|^2 + \|y\|^2 + \varepsilon\|x\|^2 + \frac{1}{\varepsilon}\|y\|^2 \tag{AM-GM}$$

$$= (1 + \varepsilon)\|x\|^2 + \left(1 + \frac{1}{\varepsilon}\right)\|y\|^2. \tag{35}$$

$\square$

### B.1. Proof of Lemma 4.2

*Proof.* For any $x, \hat{x} \in \mathcal{X}$ and for any $t$, we have

$$\langle V(\hat{x}), \hat{x} - x \rangle = \langle V(\hat{x}), \hat{x} - x_t^* \rangle$$
$$+ \langle V(\hat{x}) - V(x_t^*), x_t^* - x \rangle + \langle V(x_t^*), x_t^* - x \rangle$$
$$\leq U\|\hat{x} - x_t^*\| + L\|\hat{x} - x_t^*\|\|x_t^* - x\| + \omega_t\|x_t^*\|\|x_t^* - x\|$$
$$\leq (U + LD)\|\hat{x} - x_t^*\| + D^2\omega_t, \tag{36}$$

where we used (9) in the first inequality. As this holds for any $x \in \mathcal{X}$, we have (13). $\square$

## C. Deferred Proofs for the RG Method

### C.1. Proof of Lemma 4.1

*Proof.* As (9) holds for any $x \in \mathcal{X}$, we have the following two inequalities:

$$\langle V(x_t^*) + \omega_t x_t^*, x_t^* - x_{t+1}^* \rangle \leq 0, \quad \langle V(x_{t+1}^*) + \omega_{t+1} x_{t+1}^*, x_{t+1}^* - x_t^* \rangle \leq 0. \tag{37}$$

Combining these two inequalities yields

$$0 \geq \langle V(x_t^*) + \omega_t x_t^* - V(x_{t+1}^*) - \omega_{t+1} x_{t+1}^*, x_t^* - x_{t+1}^* \rangle \tag{38}$$

$$= \langle V(x_t^*) - V(x_{t+1}^*), x_t^* - x_{t+1}^* \rangle + \langle \omega_t x_t^* - \omega_{t+1} x_{t+1}^*, x_t^* - x_{t+1}^* \rangle \tag{39}$$

$$\geq \langle \omega_t x_t^* - \omega_{t+1} x_{t+1}^*, x_t^* - x_{t+1}^* \rangle \tag{40}$$

$$= \omega_t \|x_t^* - x_{t+1}^*\|^2 + (\omega_t - \omega_{t+1}) \langle x_{t+1}^*, x_t^* - x_{t+1}^* \rangle \tag{41}$$

$$\geq \omega_t \|x_t^* - x_{t+1}^*\|^2 - |\omega_t - \omega_{t+1}| \cdot \|x_{t+1}^*\| \cdot \|x_t^* - x_{t+1}^*\| \tag{42}$$

$$= \|x_t^* - x_{t+1}^*\| \left( \omega_t \|x_t^* - x_{t+1}^*\| - |\omega_t - \omega_{t+1}| \cdot \|x_{t+1}^*\| \right), \tag{43}$$

where the second inequality follows from the assumption that $V(x)$ is monotone (3) and the last inequality follows from the Cauchy-Schwarz inequality. It follows that

$$\|x_t^* - x_{t+1}^*\| \leq \frac{1}{\omega_t}|\omega_t - \omega_{t+1}| \cdot \|x_{t+1}^*\| \leq \left|1 - \frac{\omega_{t+1}}{\omega_t}\right| D. \tag{44}$$

$\square$

### C.2. Proof of Lemma 5.2

*Proof.* From the standard analysis of the online projected gradient descent (see, e.g., Eq.(3.3) of (Hazan, 2016)), we have

$$\|x^* - x_{t+1}\|^2 \leq \|x^* - x_t\|^2 + \|\eta_t \hat{g}_t + \gamma_t x_t\|^2 - 2 \langle \eta_t \hat{g}_t + \gamma_t x_t, x_t - x^* \rangle \tag{45}$$

for any $x^* \in \mathcal{X}$. Taking the conditional expectation given $x_t$ and letting $x^* = x_t^*$, we have

$$\mathbf{E}\left[\|x_t^* - x_{t+1}\|^2 | x_t\right] \leq \|x_t^* - x_t\|^2 + 2G^2\eta_t^2 + 2D^2\gamma_t^2 - 2 \langle \eta_t V(x_t) + \gamma_t x_t, x_t - x_t^* \rangle. \tag{46}$$

From (11), we have

$$\langle \eta_t V(x_t) + \gamma_t x_t, x_t - x_t^* \rangle = \eta_t \langle V_t(x_t), x_t - x_t^* \rangle \geq \gamma_t \| x_t - x_t^* \|^2. \tag{47}$$

Combining the above two inequalities, we obtain

$$\mathbf{E}\left[ \| x_t^* - x_{t+1} \|^2 | x_t \right] \leq (1 - 2\gamma_t) \| x_t^* - x_t \|^2 + 2G^2 \eta_t^2 + 2D^2 \gamma_t^2. \tag{48}$$

We have

$$
\begin{aligned}
\| x_t^* - x_t \|^2 &= \| x_t^* - x_{t-1}^* + x_{t-1}^* - x_t \|^2 \\
&= \| x_t^* - x_{t-1}^* \|^2 + \| x_{t-1}^* - x_t \|^2 + 2 \langle x_t^* - x_{t-1}^*, x_{t-1}^* - x_t \rangle \\
&\leq \| x_t^* - x_{t-1}^* \|^2 + \| x_{t-1}^* - x_t \|^2 + 2 \| x_t^* - x_{t-1}^* \| \| x_{t-1}^* - x_t \| && \text{(Cauchy–Schwarz)} \\
&\leq \| x_t^* - x_{t-1}^* \|^2 + \| x_{t-1}^* - x_t \|^2 + \frac{1}{\gamma_t} \| x_t^* - x_{t-1}^* \|^2 + \gamma_t \| x_{t-1}^* - x_t \|^2 && \text{(AM-GM)} \\
&\leq \left( 1 + \frac{1}{\gamma_t} \right) \| x_t^* - x_{t-1}^* \|^2 + (1 + \gamma_t) \| x_{t-1}^* - x_t \|^2.
\end{aligned}
$$

Combining this with (48), we obtain

$$\mathbf{E}\left[ \| x_t^* - x_{t+1} \|^2 | x_t \right] \leq (1 - \gamma_t) \| x_{t-1}^* - x_t \|^2 + \left( 1 + \frac{1}{\gamma_t} \right) \| x_t^* - x_{t-1}^* \|^2 + 2G^2 \eta_t^2 + 2D^2 \gamma_t^2. \tag{49}$$

Hence, $a_t := \mathbf{E}\left[ \| x_{t-1}^* - x_t \|^2 \right]$ satisfies

$$a_{t+1} \leq (1 - \gamma_t) a_t + \left( 1 + \frac{1}{\gamma_t} \right) \| x_t^* - x_{t-1}^* \|^2 + 2G^2 \eta_t^2 + 2D^2 \gamma_t^2. \tag{50}$$

$\square$

### C.3. Proof of Theorem 5.1 (Anytime Convergence)

*Proof.* Let $\gamma_t = t^{-4/5}$ and $\eta_t = (D/G) t^{-3/5}$. We first bound $\| x_t^* - x_{t+1}^* \|$. From Lemma 4.1 and Lemma B.1,

$$\| x_t^* - x_{t+1}^* \| \leq D \left| 1 - \frac{\eta_t \gamma_{t+1}}{\gamma_t \eta_{t+1}} \right| = D \left( 1 - \left( \frac{t}{t+1} \right)^{1/5} \right) \leq \frac{D}{5t}, \tag{51}$$

where the last inequality follows from the bound $(1 + 1/t)^{-1/5} \geq 1 - 1/(5t)$ for $t \geq 1$.

From (51) and Lemma 5.2, for any $t \geq 2$, we have

$$
\begin{aligned}
a_{t+1} &\leq (1 - t^{-4/5}) a_t + \left( 1 + t^{4/5} \right) \frac{D^2}{25(t-1)^2} + 2D^2 t^{-6/5} + 2D^2 t^{-8/5} \\
&\leq (1 - t^{-4/5}) a_t + \frac{D^2}{25(t-1)^2} + \frac{D^2 t^{4/5}}{25(t-1)^2} + 2D^2 t^{-6/5} + 2D^2 t^{-8/5}.
\end{aligned} \tag{52}
$$

For $t \geq 2$, we have $(t-1)^2 \geq t^2/4$, so $t^{4/5}/(t-1)^2 \leq 4t^{-6/5}$. Also, $1/(t-1)^2 \leq 4/t^2 \leq 4t^{-6/5}$ for $t \geq 2$. Therefore,

$$
\begin{aligned}
a_{t+1} &\leq (1 - t^{-4/5}) a_t + \frac{4D^2}{25} t^{-6/5} + \frac{4D^2}{25} t^{-6/5} + 2D^2 t^{-6/5} + 2D^2 t^{-8/5} \\
&\leq (1 - t^{-4/5}) a_t + 5D^2 t^{-6/5}.
\end{aligned} \tag{53}
$$

We now show by induction that $a_t \leq 10D^2 t^{-2/5}$ for $t \geq 1$.

**Base case:** As $a_t \leq D^2$ follows from the definition (since $\| x_{t-1}^* - x_t \| \leq D$), the bound $a_t \leq 10D^2 t^{-2/5}$ holds for $t = 1$ (where $10 \cdot 1^{-2/5} = 10 \geq 1$) and $t = 2$ (where $10 \cdot 2^{-2/5} \approx 7.58 \geq 1$).

**Inductive step:** Suppose that $a_t \leq 10D^2 t^{-2/5}$ holds for some $t \geq 2$. Then, (53) implies

$$
\begin{aligned}
a_{t+1} &\leq 10 \left(1 - t^{-4/5}\right) D^2 t^{-2/5} + 5D^2 t^{-6/5} \\
&= 10D^2 \left(t^{-2/5} - t^{-6/5}\right) + 5D^2 t^{-6/5} \\
&= 10D^2 t^{-2/5} - 5D^2 t^{-6/5} \\
&= 10D^2 t^{-2/5} \left(1 - \frac{1}{2} t^{-4/5}\right) \\
&= 10D^2 t^{-2/5} \left(1 - \frac{1}{2t^{4/5}}\right).
\end{aligned}
\tag{54}
$$

To complete the induction, we need to show that $t^{-2/5}(1 - 1/(2t^{4/5})) \leq (t+1)^{-2/5}$. This is equivalent to showing $(1 - 1/(2t^{4/5})) \leq (t/(t+1))^{2/5}$.

By Lemma B.1 with $\alpha = 2/5$, we have $(t/(t+1))^{2/5} \geq 1 - 2/(5t)$ for $t \geq 1$. For $t \geq 2$, we have $1/(2t^{4/5}) \leq 1/(2 \cdot 2^{4/5}) \approx 0.287 < 2/(5 \cdot 2) = 0.2...$

Actually, we verify directly: for $t \geq 2$,

$$
1 - \frac{1}{2t^{4/5}} \geq 1 - \frac{2}{5t} \quad \Leftrightarrow \quad \frac{2}{5t} \geq \frac{1}{2t^{4/5}} \quad \Leftrightarrow \quad \frac{4}{5} \geq t^{-1/5} \quad \Leftrightarrow \quad t \geq \left(\frac{5}{4}\right)^5 \approx 3.05.
\tag{55}
$$

For $t = 2$, we verify numerically: $1 - 1/(2 \cdot 2^{4/5}) \approx 0.713$ and $(2/3)^{2/5} \approx 0.858$, so the inequality holds.

Therefore, by Lemma B.1,

$$
a_{t+1} \leq 10D^2 t^{-2/5} \left(1 - \frac{2}{5t}\right) \leq 10D^2 (t+1)^{-2/5}.
\tag{56}
$$

This completes the induction, showing that $a_t \leq 10D^2 t^{-2/5}$ holds for all $t \geq 1$.

From Lemma 4.2 with $\omega_{t-1} = \gamma_{t-1}/\eta_{t-1} = G(t-1)^{-1/5}/D$, for $t \geq 2$, we have

$$
\begin{aligned}
\text{Gap}(x_t) &\leq (U + LD)\|x_t - x_{t-1}^*\| + D^2 \omega_{t-1} \\
&\leq (U + LD)\|x_t - x_{t-1}^*\| + DG(t-1)^{-1/5}.
\end{aligned}
\tag{57}
$$

Squaring and taking expectations,

$$
\begin{aligned}
\mathbf{E}[\text{Gap}(x_t)^2] &\leq 2(U + LD)^2 \, \mathbf{E}[\|x_t - x_{t-1}^*\|^2] + 2D^2 G^2 (t-1)^{-2/5} \\
&= 2(U + LD)^2 a_t + 2D^2 G^2 (t-1)^{-2/5} \\
&\leq 2(U + LD)^2 \cdot 10D^2 t^{-2/5} + 2D^2 G^2 (t-1)^{-2/5} \\
&\leq 2D^2 \left(10(U + LD)^2 + G^2\right) (t-1)^{-2/5} \\
&= O\left(D^2 (U + G + LD)^2 t^{-2/5}\right).
\end{aligned}
\tag{58}
$$

$\square$

### C.4. Proof of Theorem 5.4 (Fixed-Horizon)

*Proof of Theorem 5.4.* As $\gamma_t = \gamma_{t-1}$ and $\eta_t = \eta_{t-1}$, we have $x_t^* = x_{t-1}^*$. Hence, Lemma 5.2 implies that $a_t$ satisfies

$$
a_{t+1} \leq (1 - \gamma)a_t + 2G^2 \eta^2 + 2D^2 \gamma^2.
\tag{59}
$$

From this, we have

$$
a_T \leq (1 - \gamma)^{T-1} a_1 + \sum_{t=1}^{T-1} (1 - \gamma)^{t-1} \left(2G^2 \eta^2 + 2D^2 \gamma^2\right)
\tag{60}
$$

$$
\leq (1 - \gamma)^{T-1} D^2 + \frac{2G^2 \eta^2 + 2D^2 \gamma^2}{\gamma}.
\tag{61}
$$

From the definition of $\gamma$ and $\eta$, we have

$$a_T \leq \left(1 - \frac{\log T}{T}\right)^{T-1} D^2 + \frac{2GD^2\sqrt{\gamma}}{M} + 2D^2\gamma \lesssim D^2\left(\frac{1}{T} + \frac{G\sqrt{\gamma}}{M} + \gamma\right). \tag{62}$$

By combining this with Lemma 4.2, we obtain

$$\mathbf{E}\left[\mathrm{Gap}(x_T)^2\right] \lesssim M^2 a_T + D^4\frac{\gamma^2}{\eta^2} \tag{63}$$

$$\lesssim D^2 M\left(\frac{M}{T} + G\sqrt{\gamma} + M\gamma\right) + D^2 M G\sqrt{\gamma} \tag{64}$$

$$\lesssim D^2 M\left(\frac{M}{T} + G\sqrt{\gamma} + M\gamma\right) \tag{65}$$

$$\lesssim D^2 M\left(G\sqrt{\frac{\log T}{T}} + M\frac{\log T}{T}\right). \tag{66}$$

$\square$

## C.5. Proof of Lemma 5.5

*Proof.* From the standard analysis of the online projected gradient descent (see, e.g., Eq.(3.3) of (Hazan, 2016)), we have

$$\|x^* - x_{t+1}\|^2 \leq \|x^* - x_t\|^2 + \|\eta_t\hat{g}_t\|^2 - 2\langle\eta_t\hat{g}_t, x_t - x^*\rangle \tag{67}$$

for the unique solution $x^* \in \mathcal{X}$. Taking the conditional expectation given $x_t$, we have

$$\mathbf{E}\left[\|x^* - x_{t+1}\|^2|x_t\right] \leq \|x^* - x_t\|^2 + G^2\eta_t^2 - 2\eta_t\langle V(x_t), x_t - x^*\rangle. \tag{68}$$

From the $\lambda$-strong monotonicity of $V$ and the VI characterization of $x^*$ (i.e., $\langle V(x^*), x^* - x\rangle \leq 0$ for all $x \in \mathcal{X}$), we have

$$\langle V(x_t), x_t - x^*\rangle \geq \langle V(x^*), x_t - x^*\rangle + \lambda\|x_t - x^*\|^2 \geq \lambda\|x_t - x^*\|^2. \tag{69}$$

Combining these inequalities, we obtain

$$\mathbf{E}\left[\|x^* - x_{t+1}\|^2|x_t\right] \leq (1 - 2\eta_t\lambda)\|x^* - x_t\|^2 + G^2\eta_t^2. \tag{70}$$

Taking the full expectation, we have

$$a_{t+1} \leq (1 - 2\eta_t\lambda)a_t + G^2\eta_t^2. \tag{71}$$

$\square$

## C.6. Proof of Theorem 5.6

*Proof.* Define $C = 2D^2 + \frac{4G^2}{\lambda^2}$. We show by induction that $a_t \leq \frac{C}{t}$ for $t \geq 2$.

**Base case:** For $t = 2$, we have $a_2 \leq D^2$ by definition (since $\|x_2 - x^*\| \leq D$). By the definition of $C$, we have $C = 2D^2 + \frac{4G^2}{\lambda^2} \geq 2D^2$, so $a_2 \leq D^2 \leq \frac{C}{2}$. Hence, the bound holds for $t = 2$.

**Inductive step:** Suppose $a_t \leq \frac{C}{t}$ for some $t \geq 2$. From Lemma 5.5 with $\eta_t = (\lambda t)^{-1}$,

$$a_{t+1} \leq (1 - 2\eta_t\lambda)a_t + G^2\eta_t^2$$

$$= \left(1 - \frac{2}{\lambda t} \cdot \lambda\right)a_t + G^2\left(\frac{1}{\lambda t}\right)^2$$

$$= \left(1 - \frac{2}{t}\right)a_t + \frac{G^2}{\lambda^2 t^2}. \tag{72}$$

By the inductive hypothesis,

$$
\begin{aligned}
a_{t+1} &\leq \left(1 - \frac{2}{t}\right)\frac{C}{t} + \frac{G^2}{\lambda^2 t^2} \\
&= \frac{C}{t} - \frac{2C}{t^2} + \frac{G^2}{\lambda^2 t^2}.
\end{aligned}
\tag{73}
$$

Since $G^2/\lambda^2 \leq C/4$ by the definition of $C$, we have $\frac{G^2}{\lambda^2 t^2} \leq \frac{C}{4t^2}$. Therefore,

$$
\begin{aligned}
a_{t+1} &\leq \frac{C}{t} - \frac{2C}{t^2} + \frac{C}{4t^2} \\
&= \frac{C}{t} - \frac{3C}{2t^2} \\
&\leq \frac{C}{t} - \frac{C}{t^2} \\
&= C\left(\frac{1}{t} - \frac{1}{t^2}\right) \\
&= \frac{C(t-1)}{t^2} \\
&\leq \frac{C}{t+1},
\end{aligned}
\tag{74}
$$

where the last inequality follows from $(t-1)(t+1) = t^2 - 1 \leq t^2$, which is equivalent to $\frac{t-1}{t^2} \leq \frac{1}{t+1}$.

Hence, by induction, $a_t \leq \frac{C}{t}$ for all $t \geq 2$.

**Final bound:** From (15), we have for all $\hat{x} \in \mathcal{X}$,

$$
\mathrm{Gap}(\hat{x}) \geq \lambda\|\hat{x} - x^*\|^2,
\tag{75}
$$

which implies $\mathrm{Gap}(\hat{x}) \leq (U + LD)\|\hat{x} - x^*\|$ by the bound $\mathrm{Gap}(\hat{x}) \leq U\|\hat{x} - x^*\| + LD\|\hat{x} - x^*\|$ from Lemma 4.2 with $\omega_t = 0$.

Therefore,

$$
\begin{aligned}
\mathbf{E}[\mathrm{Gap}(x_t)^2] &\leq (U + LD)^2 \mathbf{E}[\|x_t - x^*\|^2] \\
&= (U + LD)^2 a_t \\
&\leq \frac{(U + LD)^2 C}{t} \\
&= \frac{(U + LD)^2 \left(2D^2 + \frac{4G^2}{\lambda^2}\right)}{t} \\
&= O\left(\frac{(U + LD)^2(D^2\lambda^2 + G^2)}{\lambda^2 t}\right).
\end{aligned}
\tag{76}
$$

$\square$

## D. Deferred Proofs for the ROG Method

### D.1. Proof of Lemma 6.2

*Proof.* Denote $\omega_t = \frac{\gamma_t}{\eta_t}$. As (6) can be written as $y_{t+1} = \Pi_{\mathcal{X}}(y_t - \eta_t(\hat{g}_t + \omega_t y_t))$, from the standard analysis of the online projected gradient descent, we have

$$
2\eta_t \langle \hat{g}_t + \omega_t y_t, y_{t+1} - x^* \rangle \leq \|y_t - x^*\|^2 - \|y_{t+1} - x^*\|^2 - \|y_t - y_{t+1}\|^2
\tag{77}
$$

for any $x^* \in \mathcal{X}$. Similarly, from (7), we have

$$
2\eta_t \langle \hat{g}_{t-1} + \omega_t y_t, x_t - x^* \rangle \leq \|y_t - x^*\|^2 - \|x_t - x^*\|^2 - \|y_t - x_t\|^2
\tag{78}
$$

for any $x^* \in \mathcal{X}$. Combining (77) with $x^* = x_t^*$ and (78) with $x^* = y_{t+1}$ yields

$$2\eta_t \langle \hat{g}_{t-1} - \hat{g}_t, x_t - y_{t+1} \rangle + 2\eta_t \langle \hat{g}_t + \omega_t y_t, x_t - x_t^* \rangle \tag{79}$$

$$= 2\eta_t \langle \hat{g}_{t-1} + \omega_t y_t, x_t - y_{t+1} \rangle + 2\eta_t \langle \hat{g}_t + \omega_t y_t, y_{t+1} - x_t^* \rangle \tag{80}$$

$$\leq \|y_t - y_{t+1}\|^2 - \|x_t - y_{t+1}\|^2 - \|y_t - x_t\|^2 + \|y_t - x_t^*\|^2 - \|y_{t+1} - x_t^*\|^2 - \|y_t - y_{t+1}\|^2 \tag{81}$$

$$= \|y_t - x_t^*\|^2 - \|y_{t+1} - x_t^*\|^2 - \|x_t - y_{t+1}\|^2 - \|y_t - x_t\|^2. \tag{82}$$

Rearranging yields

$$\|y_{t+1} - x_t^*\|^2 + \frac{1}{2}\|x_t - y_{t+1}\|^2$$

$$\leq 2\eta_t \langle \hat{g}_t - \hat{g}_{t-1}, x_t - y_{t+1} \rangle + 2\eta_t \langle \hat{g}_t + \omega_t y_t, x_t^* - x_t \rangle + \|y_t - x_t^*\|^2 - \frac{1}{2}\|x_t - y_{t+1}\|^2 - \|y_t - x_t\|^2$$

$$\leq 2\eta_t^2 \|\hat{g}_{t-1} - \hat{g}_t\|^2 + 2\eta_t \langle \hat{g}_t + \omega_t y_t, x_t^* - x_t \rangle + \|y_t - x_t^*\|^2 - \|y_t - x_t\|^2,$$

where the second inequality follows from $2\eta_t \langle \hat{g}_t - \hat{g}_{t-1}, x_t - y_{t+1} \rangle \leq 2\eta_t \|\hat{g}_{t-1} - \hat{g}_t\| \cdot \|x_t - y_{t+1}\| \leq 2\eta_t^2 \|\hat{g}_{t-1} - \hat{g}_t\|^2 + \frac{1}{2}\|x_t - y_{t+1}\|^2$. From the assumptions on $\hat{g}_t$ and $V(x)$, we have

$$\mathbf{E}\left[\|\hat{g}_{t-1} - \hat{g}_t\|^2\right] \leq \mathbf{E}\left[3\|V(x_{t-1}) - \hat{g}_{t-1}\|^2 + 3\|V(x_t) - \hat{g}_t\|^2 + 3\|V(x_t) - V(x_{t-1})\|^2\right] \tag{83}$$

$$\leq 6\sigma^2 + 3L^2 \mathbf{E}\left[\|x_t - x_{t-1}\|^2\right] \tag{84}$$

$$\leq 6\sigma^2 + 6L^2 \mathbf{E}\left[\|x_{t-1} - y_t\|^2 + \|x_t - y_t\|^2\right]. \tag{85}$$

Additionally, we have

$$2\mathbf{E}\left[\langle \hat{g}_t + \omega_t y_t, x_t^* - x_t \rangle | x_t, y_t\right] = 2\langle V(x_t) + \omega_t y_t, x_t^* - x_t \rangle \tag{86}$$

$$\leq 2\langle V(x_t^*) + \omega_t y_t, x_t^* - x_t \rangle \tag{87}$$

$$\leq 2\langle -\omega_t x_t^* + \omega_t y_t, x_t^* - x_t \rangle \tag{88}$$

$$= 2\omega_t \langle y_t - x_t^*, x_t^* - x_t \rangle \tag{89}$$

$$= \omega_t \left(\|x_t - y_t\|^2 - \|x_t^* - x_t\|^2 - \|x_t^* - y_t\|^2\right) \tag{90}$$

$$\leq \omega_t \left(\|x_t - y_t\|^2 - \|x_t^* - y_t\|^2\right). \tag{91}$$

Combining the preceding bounds yields

$$a_{t+1} = \mathbf{E}\left[\|y_{t+1} - x_t^*\|^2 + \frac{1}{2}\|x_t - y_{t+1}\|^2\right]$$

$$\leq \mathbf{E}\left[(1 - \gamma_t)\|y_t - x_t^*\|^2 + 12L^2\eta_t^2\|x_{t-1} - y_t\|^2 + \left(12L^2\eta_t^2 + \gamma_t - 1\right)\|x_t - y_t\|^2 + 12\sigma^2\eta_t^2\right]$$

$$\leq \mathbf{E}\left[(1 - \gamma_t)\|y_t - x_t^*\|^2 + 12L^2\eta_t^2\|x_{t-1} - y_t\|^2 + 12\sigma^2\eta_t^2\right]$$

$$\leq \mathbf{E}\left[\left(1 - \frac{\gamma_t}{2}\right)\|y_t - x_{t-1}^*\|^2 + 12L^2\eta_t^2\|x_{t-1} - y_t\|^2 + \left(1 + \frac{2}{\gamma_t}\right)\|x_t^* - x_{t-1}^*\|^2 + 12\sigma^2\eta_t^2\right]$$

$$\leq \left(1 - \frac{\gamma_t}{2}\right)a_t + \left(1 + \frac{2}{\gamma_t}\right)\|x_t^* - x_{t-1}^*\|^2 + 12\sigma^2\eta_t^2,$$

where the third inequality follows from Lemma B.2 with $\varepsilon = \gamma_t/2 \leq 1/2$ and the last inequality follows from $\eta_t \leq 1/(6L)$. □

## D.2. Proof of Theorem 6.1 (Anytime Convergence)

*Proof.* As (17) is clear for $t \leq 4$, we assume that $t \geq 5$ in the following. Denote $M = U + LD$. From Lemma 4.2, we have

$$\mathbf{E}\left[\text{Gap}(x_t)^2\right] \leq \mathbf{E}\left[2M^2\|x_t - x_t^*\|^2 + 2D^4\omega_t^2\right] \leq 8M^2 a_{t+1} + 2D^4\omega_t^2. \tag{92}$$

Define $t_0 := (DL/\sigma)^{4/3}$. For $t \leq t_0$, we use:

$$\eta_t = \frac{1}{6L}, \quad \gamma_t = t^{-1/2}, \quad \omega_t = 6Lt^{-1/2}. \tag{93}$$

For $t > t_0$, we use:

$$\eta_t = \frac{D^{4/5}}{6L^{1/5}\sigma^{4/5}t^{3/5}}, \quad \gamma_t = \frac{(DL)^{2/5}}{\sigma^{2/5}t^{4/5}}, \quad \omega_t = \frac{6L^{3/5}\sigma^{2/5}}{D^{2/5}t^{1/5}}. \tag{94}$$

We now bound $\|x_t^* - x_{t+1}^*\|$ using Lemma 4.1. For $t \leq t_0$, we have

$$\frac{\omega_{t+1}}{\omega_t} = \frac{6L(t+1)^{-1/2}}{6Lt^{-1/2}} = \left(\frac{t}{t+1}\right)^{1/2}. \tag{95}$$

By Lemma B.1 with $\alpha = 1/2$, we have $(t/(t+1))^{1/2} \geq 1 - 1/(2t)$, so

$$\|x_t^* - x_{t+1}^*\| \leq \left|1 - \frac{\omega_{t+1}}{\omega_t}\right| D \leq \frac{D}{2t}. \tag{96}$$

For $t > t_0$, we have

$$\frac{\omega_{t+1}}{\omega_t} = \frac{6L^{3/5}\sigma^{2/5}D^{-2/5}(t+1)^{-1/5}}{6L^{3/5}\sigma^{2/5}D^{-2/5}t^{-1/5}} = \left(\frac{t}{t+1}\right)^{1/5}. \tag{97}$$

By Lemma B.1 with $\alpha = 1/5$, we have $(t/(t+1))^{1/5} \geq 1 - 1/(5t)$, so

$$\|x_t^* - x_{t+1}^*\| \leq \left|1 - \frac{\omega_{t+1}}{\omega_t}\right| D \leq \frac{D}{5t}. \tag{98}$$

We will establish the following bounds by applying Lemma 6.2 with the above parameter choices:

$$a_t \leq \frac{24D^2}{t} \quad \text{for } 5 \leq t \leq t_0, \tag{99}$$

$$a_t \leq \frac{24D^2}{t_0^{3/5}t^{2/5}} \quad \text{for } t \geq t_0. \tag{100}$$

**Case $t \leq t_0$:** We show (99) by induction. Substituting $\gamma_t = t^{-1/2}$, $\eta_t = (6L)^{-1}$, and $\|x_t^* - x_{t-1}^*\| \leq D/(2(t-1))$ into Lemma 6.2 and using $(t-1)^2 \geq t^2/4$ for $t \geq 2$, we obtain

$$a_{t+1} \leq \left(1 - \frac{1}{2\sqrt{t}}\right) a_t + \frac{(1 + 2\sqrt{t})D^2}{t^2} + \frac{\sigma^2}{3L^2}. \tag{101}$$

We now bound the error terms. For the first error term, since $t \geq 5$ implies $t^{1/2} \geq 2$, we have $1/t^2 \leq 1/(2t^{3/2})$, and thus

$$\frac{(1 + 2\sqrt{t})D^2}{t^2} = \frac{D^2}{t^2} + \frac{2D^2}{t^{3/2}} \leq \frac{D^2}{2t^{3/2}} + \frac{2D^2}{t^{3/2}} = \frac{5D^2}{2t^{3/2}}. \tag{102}$$

For the noise term, since $t \leq t_0 = (DL/\sigma)^{4/3}$ implies $\sigma/(DL) \leq t^{-3/4}$, we have $\sigma^2/(D^2L^2) \leq t^{-3/2}$, and thus

$$\frac{\sigma^2}{3L^2} \leq \frac{D^2}{3t^{3/2}}. \tag{103}$$

Combining these bounds, we obtain

$$a_{t+1} \leq \left(1 - \frac{1}{2\sqrt{t}}\right) a_t + \frac{5D^2}{2t^{3/2}} + \frac{D^2}{3t^{3/2}} \leq \left(1 - \frac{1}{2\sqrt{t}}\right) a_t + \frac{3D^2}{t^{3/2}}. \tag{104}$$

From the definition of $a_t$, we have $a_5 \leq \frac{3}{2}D^2$ (since $\|y_t - x_{t-1}^*\| \leq D$ and $\|x_{t-1} - y_t\| \leq D$). Since $\frac{24D^2}{5} = 4.8D^2 > \frac{3}{2}D^2$, the bound (99) holds for $t = 5$.

Suppose (99) holds for some $t \geq 5$. Then, from (104), we have

$$a_{t+1} \leq 24 \left(1 - \frac{1}{2\sqrt{t}}\right) \frac{D^2}{t} + \frac{3D^2}{t^{3/2}} = 24 \frac{D^2}{t}\left(1 - \frac{3}{8\sqrt{t}}\right) \leq 24\frac{D^2}{t}\frac{t}{t+1} = 24\frac{D^2}{t+1}, \tag{105}$$

where the last inequality follows from $\frac{1}{t+1} \leq \frac{3}{8\sqrt{t}}$ for $t \geq 5$. This completes the induction, establishing (99) for $5 \leq t \leq t_0$.

**Case $t \geq t_0$:** We show (100) by induction. For $t > t_0$, substituting the parameter values and $\|x_t^* - x_{t-1}^*\| \leq D/(5(t-1))$ into Lemma 6.2, we obtain

$$a_{t+1} \leq \left(1 - \frac{t_0^{3/10}}{2t^{4/5}}\right) a_t + \frac{2D^2}{t_0^{3/10}t^{6/5}}, \tag{106}$$

where we used $\gamma_t = (DL)^{2/5}\sigma^{-2/5}t^{-4/5} = t_0^{3/10}t^{-4/5}$ (since $t_0^{3/10} = (DL/\sigma)^{2/5}$) and verified that the noise term $12\sigma^2\eta_t^2$ is dominated by the other terms for $t > t_0$.

As $a_t \leq \frac{3}{2}D^2$ follows from the definition of $a_t$, (100) holds for $t \leq 2^{10}t_0^{-3/2}$. In addition, if $t_0 \geq 4$ and $t \leq t_0$, as (99) holds for any $t \in [4, t_0]$, we have

$$a_{\lfloor t_0 \rfloor} \leq 6\frac{D^2}{\lfloor t_0 \rfloor} \leq 24\frac{D^2}{t_0^{3/5}\lfloor t_0 \rfloor^{2/5}}. \tag{107}$$

Hence, (100) holds for $t = t_1 := \lfloor \max\{2^{10}t_0^{-3/2}, t_0\}\rfloor \geq 4$. If (100) holds for some $t \geq t_1$, (106) implies that

$$a_{t+1} \leq \left(1 - \frac{t_0^{3/10}}{2t^{4/5}}\right) \cdot \frac{24}{t_0^{3/5}t^{2/5}} + 2\frac{D^2}{t_0^{3/10}t^{6/5}} \tag{108}$$

$$= 24\frac{D^2}{t_0^{3/5}}\left(\frac{1}{t^{2/5}} - \frac{5}{12}\cdot\frac{t_0^{3/10}}{t^{6/5}}\right) = 24\frac{D^2}{t_0^{3/5}}\left(\frac{1}{t^{2/5}} - \frac{5}{12}\cdot\frac{(t_0^{3/2}t)^{1/5}}{t^{7/5}}\right). \tag{109}$$

When $t_0 \geq 4$, as we have $t \geq t_1 \geq 4$, we have $t_0^{3/2}t \geq 2^5$. When $t_0 < 4$, as we have $t \geq \lfloor 2^{10}t_0^{-3/2}\rfloor \geq 2^9 t_0^{-3/2}$, we have $t_0^{3/2}t \geq 2^9$. Consequently, as $t_0^{3/2}t \geq 2^5$ for $t \geq t_1$, we have

$$a_{t+1} \leq 24\frac{D^2}{t_0^{3/5}}\left(\frac{1}{t^{2/5}} - \frac{5}{12}\cdot\frac{2}{t^{7/5}}\right) \leq 24\frac{D^2}{t_0^{3/5}}\left(\frac{1}{t^{2/5}} - \frac{2}{5}\cdot\frac{1}{t^{7/5}}\right) \leq 24\frac{D^2}{t_0^{3/5}(t+1)^{2/5}}, \tag{110}$$

where the last inequality follows from Lemma B.1 with $\alpha = 2/5$.

Combining the bounds on $a_t$ with (92), we obtain

$$\mathbf{E}\left[\text{Gap}(x_t)^2\right] = O(M^2 a_{t+1} + D^4\omega_t^2)$$

$$= O\left(D^2(U + LD)^2 \max\left\{t^{-1}, t_0^{-3/5}t^{-2/5}\right\}\right). \tag{111}$$

Since $t_0 = (DL/\sigma)^{4/3}$, we have $t_0^{-3/5} = (\sigma/(DL))^{4/5}$, yielding the stated bound. $\qquad\square$

### D.3. Proof of Theorem 6.3 (Fixed-Horizon)

*Proof of Theorem 6.3.* As $\gamma_t = \gamma_{t-1}$ and $\eta_t = \eta_{t-1}$, we have $x_t^* = x_{t-1}^*$. Hence, Lemma 6.2 implies that $a_t$ satisfies

$$a_{t+1} \leq \left(1 - \frac{1}{2}\gamma\right) a_t + 12\sigma^2\eta^2. \tag{112}$$

This recursion and the definition of $\gamma$ leads to

$$a_T \leq \left(1 - \frac{1}{2}\gamma\right)^{T-1} a_1 + 12\sigma^2\eta^2 \sum_{t=1}^{T-1}\left(1 - \frac{1}{2}\gamma\right)^{t-1} \tag{113}$$

$$\leq \left(1 - 2\frac{\log T}{T}\right)^{T-1} a_1 + 12\frac{\sigma^2\eta^2}{\gamma} \lesssim \frac{D^2}{T^2} + \frac{12\sigma^2\eta^2}{\gamma}. \tag{114}$$

Combining this with Lemma 4.2 and the definitions of $\eta$ and $\gamma$, we obtain

$$\mathbf{E}\left[\mathrm{Gap}_T(x_T)^2\right] \lesssim M^2 a_T + D^4 \frac{\gamma^2}{\eta^2} \tag{115}$$

$$\lesssim M^2 \left(\frac{D^2}{T^2} + \frac{\sigma^2 \eta^2}{\gamma}\right) + D^4 \frac{\gamma^2}{\eta^2} \tag{116}$$

$$\leq M^2 \left(\frac{D^2}{T^2} + \frac{D^2 \sigma \gamma^{3/2}}{M \gamma}\right) + D^4 \gamma^2 \left(\frac{M\sigma}{D^2 \gamma^{3/2}} + 36 L^2\right) \tag{117}$$

$$= D^2 \left(2 M \sigma \sqrt{\gamma} + \frac{M^2}{T^2} + 36 D^2 L^2 \gamma^2\right) \tag{118}$$

$$\lesssim D^2 \left((U + DL)\sigma \sqrt{\frac{\log T}{T}} + \frac{1}{T^2}(U + DL \log T)^2\right). \tag{119}$$

$\square$

## D.4. Proof of Lemma 6.4

*Proof.* From the standard analysis of the online projected gradient descent applied to (25), we have

$$2\eta_t \langle \hat{g}_t, y_{t+1} - x^* \rangle \leq \|y_t - x^*\|^2 - \|y_{t+1} - x^*\|^2 - \|y_t - y_{t+1}\|^2. \tag{120}$$

Similarly, from (26), we have

$$2\eta_t \langle \hat{g}_{t-1}, x_t - x^* \rangle \leq \|y_t - x^*\|^2 - \|x_t - x^*\|^2 - \|y_t - x_t\|^2 \tag{121}$$

(here we used that $\eta_t = \eta_{t-1}$ is not required; the bound holds for any comparator). Combining (120) and (121) with comparator $y_{t+1}$ yields

$$2\eta_t \langle \hat{g}_{t-1} - \hat{g}_t, x_t - y_{t+1} \rangle + 2\eta_t \langle \hat{g}_t, x_t - x^* \rangle \tag{122}$$

$$\leq \|y_t - x^*\|^2 - \|y_{t+1} - x^*\|^2 - \|x_t - y_{t+1}\|^2 - \|y_t - x_t\|^2. \tag{123}$$

Rearranging yields

$$\|y_{t+1} - x^*\|^2 + \frac{1}{2}\|x_t - y_{t+1}\|^2$$

$$\leq 2\eta_t \langle \hat{g}_t - \hat{g}_{t-1}, x_t - y_{t+1} \rangle + 2\eta_t \langle \hat{g}_t, x^* - x_t \rangle + \|y_t - x^*\|^2 - \frac{1}{2}\|x_t - y_{t+1}\|^2 - \|y_t - x_t\|^2$$

$$\leq 2\eta_t^2 \|\hat{g}_{t-1} - \hat{g}_t\|^2 + 2\eta_t \langle \hat{g}_t, x^* - x_t \rangle + \|y_t - x^*\|^2 - \|y_t - x_t\|^2,$$

where the second inequality follows from $2\eta_t \langle \hat{g}_t - \hat{g}_{t-1}, x_t - y_{t+1} \rangle \leq 2\eta_t^2 \|\hat{g}_{t-1} - \hat{g}_t\|^2 + \frac{1}{2}\|x_t - y_{t+1}\|^2$. From the assumptions on $\hat{g}_t$ and $V(x)$, we have

$$\mathbf{E}\left[\|\hat{g}_{t-1} - \hat{g}_t\|^2\right] \leq 6\sigma^2 + 6L^2 \mathbf{E}\left[\|x_{t-1} - y_t\|^2 + \|x_t - y_t\|^2\right]. \tag{124}$$

By the strong monotonicity of $V$,

$$2\mathbf{E}\left[\langle \hat{g}_t, x^* - x_t \rangle | x_t\right] = 2\langle V(x_t), x^* - x_t \rangle \tag{125}$$

$$\leq 2\langle V(x^*), x^* - x_t \rangle - 2\lambda \|x_t - x^*\|^2 \tag{126}$$

$$\leq -2\lambda \|x_t - x^*\|^2, \tag{127}$$

where the last inequality follows from the VI characterization of $x^*$, i.e., $\langle V(x^*), x^* - x \rangle \leq 0$ for all $x \in \mathcal{X}$.

Furthermore,

$$\|x_t - x^*\|^2 = \|x_t - y_t + y_t - x^*\|^2 \geq \frac{1}{2}\|y_t - x^*\|^2 - \|x_t - y_t\|^2. \tag{128}$$

Combining the preceding bounds yields

$$
\begin{aligned}
a_{t+1} &= \mathbf{E}\left[\|y_{t+1} - x^*\|^2 + \frac{1}{2}\|x_t - y_{t+1}\|^2\right] \\
&\leq \mathbf{E}\left[(1 - \eta_t\lambda)\|y_t - x^*\|^2 + 12L^2\eta_t^2\|x_{t-1} - y_t\|^2 + \left(12L^2\eta_t^2 + 2\eta_t\lambda - 1\right)\|x_t - y_t\|^2 + 12\sigma^2\eta_t^2\right] \\
&\leq \mathbf{E}\left[(1 - \eta_t\lambda)\|y_t - x^*\|^2 + 12L^2\eta_t^2\|x_{t-1} - y_t\|^2 + 12\sigma^2\eta_t^2\right] \\
&\leq (1 - \eta_t\lambda)a_t + 12\sigma^2\eta_t^2,
\end{aligned}
$$

where the second inequality uses $12L^2\eta_t^2 + 2\eta_t\lambda - 1 \leq 0$ (which holds when $\eta_t \leq 1/(6L)$ and $\lambda \leq L$), and the last inequality uses $\eta_t \leq 1/(6L)$. $\qquad\square$

### D.5. Proof of Theorem 6.5

*Proof.* Note that $\eta_t = \frac{c}{\lambda(t+t_0)} \leq \frac{c}{\lambda t_0} \leq \frac{1}{6L}$ for all $t \geq 1$. From Lemma 6.4, we have

$$
a_{t+1} \leq \left(1 - \frac{c}{t + t_0}\right)a_t + \frac{12c^2\sigma^2}{\lambda^2(t + t_0)^2}. \tag{129}
$$

Let $s = t + t_0$ and $b_s = a_{s-t_0} = a_t$. Then the recursion becomes

$$
b_{s+1} \leq \left(1 - \frac{c}{s}\right)b_s + \frac{12c^2\sigma^2}{\lambda^2 s^2} \tag{130}
$$

for $s \geq t_0 + 1$, with initial condition $b_{t_0+1} = a_1 \leq D^2$.

Define $A = \frac{12c^2\sigma^2}{\lambda^2(c-1)}$ and $B = D^2(t_0 + 1)^c$. We show by induction that $b_s \leq \frac{A}{s} + \frac{B}{s^c}$ for all $s \geq t_0 + 1$.

For the base case $s = t_0 + 1$, we have $\frac{A}{t_0+1} + \frac{B}{(t_0+1)^c} = \frac{A}{t_0+1} + D^2 \geq D^2 \geq b_{t_0+1}$.

For the inductive step, assume $b_s \leq \frac{A}{s} + \frac{B}{s^c}$ for some $s \geq t_0 + 1$. From (130),

$$
b_{s+1} \leq \left(1 - \frac{c}{s}\right)\left(\frac{A}{s} + \frac{B}{s^c}\right) + \frac{A(c-1)}{s^2} \tag{131}
$$

$$
= \frac{A(s-c)}{s^2} + \frac{A(c-1)}{s^2} + \frac{B}{s^c}\left(1 - \frac{c}{s}\right) = \frac{A}{s}\left(1 - \frac{1}{s}\right) + \frac{B}{s^c}\left(1 - \frac{c}{s}\right). \tag{132}
$$

By Lemma B.1, $1 - \frac{1}{s} \leq \frac{s}{s+1}$ and $1 - \frac{c}{s} \leq \left(\frac{s}{s+1}\right)^c$. Therefore, $b_{s+1} \leq \frac{A}{s+1} + \frac{B}{(s+1)^c}$.

Since $s = t + t_0 \geq t$, we have

$$
a_t = b_{t+t_0} \leq \frac{A}{t + t_0} + \frac{B}{(t + t_0)^c} = O\left(\frac{c\sigma^2}{\lambda^2(t_0 + t)} + \frac{D^2}{(1 + t/t_0)^c}\right). \tag{133}
$$

From (15), we have $\mathrm{Gap}(x_t) \leq (U + LD)\|x_t - x^*\|$. Since $\|x_t - x^*\|^2 \leq 2\|y_t - x^*\|^2 + 2\|x_t - y_t\|^2 \leq 4a_t + 4a_{t+1}$, we have

$$
\mathbf{E}\left[\mathrm{Gap}(x_t)^2\right] \leq (U + LD)^2\,\mathbf{E}\left[\|x_t - x^*\|^2\right] = O\left((U + LD)^2\left(\frac{c\sigma^2}{\lambda^2(t_0 + t)} + \frac{D^2}{(1 + t/t_0)^c}\right)\right). \tag{134}
$$

$\qquad\square$

## E. Proofs for Regret Analysis

*Proof of Proposition 7.1.* The update $x_{t+1} = \Pi_{\mathcal{X}}((1 - \gamma_t)x_t - \eta_t\hat{g}_t)$ can be written as $x_{t+1} = \Pi_{\mathcal{X}}(x_t - \eta_t(\hat{g}_t + \omega_t x_t))$, where $\omega_t = \gamma_t/\eta_t$. By the standard projected gradient descent analysis,

$$
\begin{aligned}
&\langle \hat{g}_t + \omega_t x_t, x_t - u\rangle \\
&\leq \frac{\|x_t - u\|^2 - \|x_{t+1} - u\|^2}{2\eta_t} + \frac{\eta_t}{2}\|\hat{g}_t + \omega_t x_t\|^2.
\end{aligned}
$$

Since $\mathbf{E}[\hat{g}_t|x_t] = V(x_t)$ and $\mathbf{E}[\|\hat{g}_t\|^2|x_t] \leq G^2$, taking conditional expectation gives

$$\mathbf{E}[\langle V(x_t) + \omega_t x_t, x_t - u\rangle \,|x_t]$$
$$\leq \frac{\|x_t - u\|^2 - \mathbf{E}[\|x_{t+1} - u\|^2|x_t]}{2\eta_t} + \eta_t G^2 + \eta_t \omega_t^2 D^2.$$

Using $\langle x_t, x_t - u\rangle \geq -D^2$ (which follows from $\langle x_t, x_t - u\rangle = \|x_t\|^2 - \langle x_t, u\rangle \geq -\|x_t\|\|u\| \geq -D^2$), we obtain

$$\mathbf{E}[\langle V(x_t), x_t - u\rangle \,|x_t]$$
$$\leq \frac{\|x_t - u\|^2 - \mathbf{E}[\|x_{t+1} - u\|^2|x_t]}{2\eta_t} + \eta_t G^2 + \omega_t D^2.$$

Summing over $t = 1, \ldots, T$ and taking full expectation,

$$\mathbf{E}[\mathrm{Reg}_T(u)] \leq \sum_{t=1}^{T} \frac{\mathbf{E}[\|x_t - u\|^2] - \mathbf{E}[\|x_{t+1} - u\|^2]}{2\eta_t}$$
$$+ \sum_{t=1}^{T} \eta_t G^2 + \sum_{t=1}^{T} \omega_t D^2.$$

For non-increasing $\{\eta_t\}$, the telescoping sum satisfies

$$\sum_{t=1}^{T} \frac{\mathbf{E}[\|x_t - u\|^2] - \mathbf{E}[\|x_{t+1} - u\|^2]}{2\eta_t}$$
$$= \frac{\mathbf{E}[\|x_1 - u\|^2]}{2\eta_1} + \sum_{t=2}^{T} \mathbf{E}[\|x_t - u\|^2]\left(\frac{1}{2\eta_t} - \frac{1}{2\eta_{t-1}}\right)$$
$$- \frac{\mathbf{E}[\|x_{T+1} - u\|^2]}{2\eta_T}$$
$$\leq \frac{D^2}{2\eta_1} + \sum_{t=2}^{T} D^2\left(\frac{1}{2\eta_t} - \frac{1}{2\eta_{t-1}}\right) = \frac{D^2}{2\eta_T}.$$

This completes the proof. $\qquad\qquad\square$

## F. Numerical Experiments

We validate the theoretical rates on synthetic monotone and strongly monotone VIs. We construct a linear operator $V(x) = Ax + b$ on $\mathcal{X} = [-1, 1]^d$ with $d = 10$, where $A = S + K + \mu I$ is the sum of a symmetric positive semidefinite matrix $S$, a skew-symmetric matrix $K$, and a strong monotonicity shift $\mu I$. The monotone case uses $\mu = 0$, and the strongly monotone case uses $\mu > 0$. Noisy feedback is generated as $\hat{g}_t = V(x_t) + \xi_t$ with $\xi_t \sim \mathcal{N}(0, \sigma^2 I)$. All experiments use horizon $T = 500000$ and report averages over five random seeds. For the strongly monotone case, we set $\mu = 0.2$ and use $c = 2$ in the step-size schedule.

For the monotone case, we use the anytime schedules in Theorem 5.1 and Theorem 6.1. For the strongly monotone case, we use the theorem-matched schedules $\gamma_t = 0$, $\eta_t = 1/(\lambda t)$ for RG, and $\eta_t = c/(\lambda(t + t_0))$ for ROG, with $t_0 = \lceil 6cL/\lambda \rceil$ and $c = 2$. We report the squared gap $\mathrm{Gap}(x_t)^2$ on a log-log scale.

The extended horizon of $T = 500000$ iterations reveals clear asymptotic behavior that closely aligns with the theoretical predictions. In the monotone settings, both RG and ROG exhibit convergence rates consistent with the $t^{-0.4}$ reference line. When noise is high ($\sigma = 0.1$), RG and ROG perform similarly. However, as noise decreases, ROG's variance-adaptive behavior becomes increasingly apparent: at $\sigma = 0.01$, ROG achieves roughly one order of magnitude better convergence than RG, and in the noiseless case ($\sigma = 0$), ROG approaches the faster $t^{-1}$ rate while RG remains bounded by $t^{-0.4}$. This dramatic improvement in the low-noise regime confirms the variance-adaptive term in Theorem 6.1.

In the strongly monotone settings, both methods exhibit the predicted $t^{-1}$ convergence rate. At high noise ($\sigma = 0.1$), RG and ROG perform comparably. At moderate noise ($\sigma = 0.01$), ROG achieves convergence levels several orders of

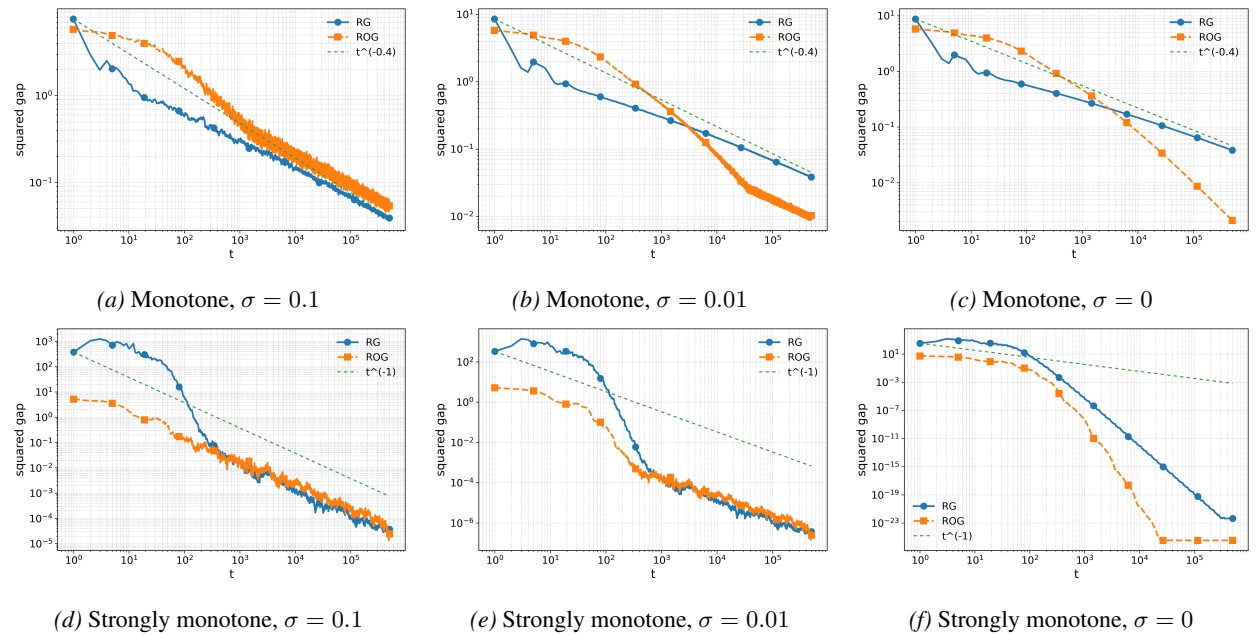

*Figure 1.* Synthetic VI experiments over $T = 500000$ iterations. RG uses solid lines with circle markers and ROG uses dashed lines with square markers; reference slopes are shown as dashed green lines. In the monotone setting, the reference slope corresponds to $t^{-0.4}$ (equivalent to $t^{-2/5}$), while in the strongly monotone setting it corresponds to $t^{-1}$, matching the dominant terms in Theorems 5.1–6.1 and Theorems 5.6–6.5.

magnitude better than RG. Most strikingly, in the noiseless case ($\sigma = 0$), ROG converges to machine precision (below $10^{-25}$), while RG reaches approximately $10^{-23}$. This exceptionally fast convergence of ROG demonstrates the $t^{-c}$ term in Theorem 6.5, which allows arbitrarily fast decay for any constant $c > 1$.

Overall, the long-horizon experiments confirm that the theoretical rates accurately capture the asymptotic behavior of both methods. The results highlight the significant practical advantage of optimism, particularly in low-noise regimes where ROG can achieve convergence rates multiple orders of magnitude better than RG. The source code for reproducing these experiments is provided in the supplementary material.

## G. Additional Remarks

*Remark* G.1 (Difficulty of extending bandit lower bounds). The lower bound construction of Fiegel et al. (2025) for bandit feedback relies on the following key observation. Consider two game matrices that differ by a perturbation of magnitude $\varepsilon$. When the algorithm outputs a $\delta_t$-approximate solution at round $t$, the Kullback–Leibler (KL) divergence between the bandit feedback distributions under the two games is of order $(\delta_t \varepsilon)^2$. Since both $\delta_t$ and $\varepsilon$ can be made small, distinguishing between the two games requires many rounds, which limits the achievable convergence rate.

In contrast, under stochastic gradient feedback, the player observes a noisy gradient $\hat{g}_t = V(x_t) + \xi_t$ rather than a scalar loss value. For the same pair of perturbed games, the KL divergence between the gradient feedback distributions becomes $O(\varepsilon^2)$, independent of the approximation quality $\delta_t$. This loss of the $\delta_t$ factor in the KL bound prevents a direct application of the same information-theoretic argument, suggesting that establishing lower bounds for the stochastic gradient setting may require fundamentally different techniques.

