# OpenReview forum: "Last-Iterate Convergence of Regularized Gradient Methods for Stochastic Monotone Variational Inequalities"
_ICML.cc/2026/Conference — ICML 2026 regular_

### Official Review · Reviewer_KeEa · 2026-02-26

**Soundness:** 4
**Presentation:** 3
**Significance:** 4
**Originality:** 4
**Overall Recommendation:** 6
**Confidence:** 4

**Summary:**

## Anytime Last-Iterate Convergence for Stochastic Monotone VIs

This paper studies **anytime last-iterate convergence** of first-order methods for **stochastic smooth monotone variational inequalities (VIs)**.

The problem class captures:

- Convex–concave saddle-point problems
- Nash equilibrium computation in monotone games
- Settings with stochastic gradient feedback

While **average-iterate convergence** is well understood in such settings, **anytime last-iterate guarantees** remain limited, especially for uncoupled dynamics.

---

## Algorithms Analyzed

The authors analyze two algorithms:

1. **Regularized Gradient (RG)**
2. **Regularized Optimistic Gradient (ROG)**

Both methods introduce **time-varying regularization** via a shrinkage term.
This effectively converts the original monotone VI into a sequence of **strongly monotone surrogate problems**.

---

## Performance Metric

The analysis tracks how close the iterate \( x_t \) is to a solution of the unregularized VI using the gap function:

$$
\mathrm{Gap}(x_t)
$$
$$
=\max_{x \in \mathcal{X}}
\langle V(x_t), x_t - x \rangle.
$$

The authors establish convergence of

$$
\mathbb{E}[\mathrm{Gap}(x_t)^2] \to 0
$$

under appropriate parameter schedules.



## Convergence Rates

### General Monotone VIs

- **RG** achieves an anytime rate
  $$
  O(t^{-2/5}).
  $$

- **ROG** achieves a variance-adaptive rate
  $$
  O(\sigma^{4/5} t^{-2/5} + t^{-1}),
  $$
  interpolating between:
  - $O(t^{-1})$ in the low-noise regime
  - $O(\sigma^{4/5} t^{-2/5})$ in the high-noise regime

### Strongly Monotone Case

Both methods achieve:

$$
O(1/t)
$$

type rates.



## Additional Results

The paper also provides:

- Fixed-horizon guarantees for both algorithms
- A regret bound for the RG method

**Compliance With Llm Reviewing Policy:**

Affirmed.

**Key Questions For Authors:**

none

**Limitations:**

### Additional Comments

 **Introduction:**

$$
\bar{x}_T
$$

does not need to store all past iterates, since

  $$
  \bar{x}_{T+1}=
$$

$ \frac{T}{T+1} \bar{x}_T + \frac{1}{T+1} x_{T+1}.$
  Moreover, players can commit to such an action rule.

- **Contribution:**
  Highlight early that $\gamma_t \overset{t \rightarrow \infty}{\rightarrow} 0$ and $\omega_t \rightarrow 0$. Consequently, $x_t^* \rightarrow x^*$, and convergence results are established for the original problem ($V$), not merely for the surrogate (regularized) problem ($V_t$).

- **Contribution:**
  The statement about a “fundamental gap between $E[\mathrm{Gap}^2]$ and $E[\mathrm{Gap}]^2$” should clarify explicitly what that gap is and why it matters.

- **Analysis Overview:**
  Specify “regularized **monotone** VI.” Note that $V_t$ is strongly monotone only if $V$ is monotone.

- **Notation:**
  The existence of $U$ such that $\|V(x)\| \le U$ is implied by Lipschitz continuity on a compact set. This could be reformulated accordingly.

- **Assumption:**
  The statement “W.l.o.g. we assume $\lambda \le L$” follows from $L$-Lipschitz continuity and $\lambda$-strong monotonicity via Cauchy–Schwarz; it may not require a separate assumption.

**Strengths And Weaknesses:**

### Soundness

**Strengths:**
The technical development is coherent and well-structured. The regularized-operator framework provides a clean mechanism for deriving contraction-style recursions even in the absence of strong monotonicity. The main recursive inequalities are carefully derived, and the parameter choices appear to balance contraction, bias drift, and stochastic noise in a principled way. The exponent arithmetic underlying the $t^{-2/5}$ and variance-adaptive rates is internally consistent.

The strongly monotone analysis follows standard stochastic approximation techniques and appears correct under the stated assumptions. The variance-adaptive schedule for ROG is particularly well motivated and technically nontrivial.

**Weaknesses:**
The most delicate part of the analysis concerns the recursion for the ROG method, where optimism and regularization interact. While the proof sketches are plausible, the correctness depends critically on careful control of gradient difference terms and stability of the regularized solutions. These steps rely heavily on appendix arguments and would benefit from additional intuition or intermediate lemmas in the main text.

The monotone-case rate $O(t^{-2/5})$ is slower than classical $O(1/\sqrt{t})$ average-iterate rates, and the paper does not provide matching lower bounds to justify whether this exponent is intrinsic or an artifact of the proof technique.

Overall, I find the paper technically sound based on the arguments presented.

---

### Presentation

**Strengths:**
The paper is clearly written and logically organized. The high-level motivation—contrasting average-iterate and last-iterate guarantees—is well articulated. The regularization framework is explained clearly, and the role of each parameter in the bias–variance tradeoff is easy to follow. The comparison table summarizing rates is helpful.

**Weaknesses:**
Some of the more technical recursions are presented in compressed form, and readers may need to rely heavily on the appendix to verify details. Including a more explicit explanation of how the exponent $2/5$ arises from balancing terms would improve accessibility. The discussion of optimality and comparison to possible lower bounds could also be expanded.

Overall, the presentation is strong for a theory paper, though somewhat dense in the ROG section.

---

> ### Author Rebuttal · Authors · 2026-03-30
>
> We are grateful to Reviewer KeEa for the detailed and constructive comments. We respond to each point below.
>
> ### On the Monotone-Case Rate $O(t^{-2/5})$ vs. Classical $O(1/\sqrt{t})$
>
> The comparison requires careful attention to the convergence metric:
>
> - **Our result**: $\mathbb{E}[\text{Gap}(x\_t)^2] = O(t^{-2/5})$, which implies $\mathbb{E}[\text{Gap}(x\_t)] = O(t^{-1/5})$ by Jensen's inequality. This is a last-iterate guarantee.
> - **Classical $O(1/\sqrt{t})$**: This rate is for the average iterate $\bar{x}\_t$, not the last iterate. Specifically, $\text{Gap}(\bar{x}\_t) = O(1/\sqrt{t})$ follows from standard online learning / mirror descent analysis.
>
> To our knowledge, no prior work gives an $O(1/\sqrt{t})$ rate for the last iterate in stochastic monotone VIs in terms of $\mathbb{E}[\text{Gap}(x\_t)]$. The closest prior last-iterate guarantees are due to Abe et al. (2024, 2025), but these are fixed-horizon results with rates $O(T^{-1/10})$ and $O(T^{-1/7})$. Our bound is instead anytime, and yields $O(t^{-1/5})$.
>
> Whether the $t^{-2/5}$ rate for $\mathbb{E}[\text{Gap}(x\_t)^2]$ can be improved to match the average-iterate rate remains an open question. As discussed in Remark D.1, establishing lower bounds for last-iterate convergence under stochastic gradient feedback requires techniques beyond those used in the bandit setting, because the KL divergence between gradient distributions loses the $\delta\_t$ dependence that is used in the bandit lower bound. We will expand this discussion in the revised version.
>
> ### On Highlighting Early Decay of $\eta\_t$
>
> We agree that the role of $\eta\_t$ in the early phase of ROG (where $\eta\_t$ is constant at $1/(6L)$) deserves more emphasis. In this phase, the constant step size combined with strong regularization decay ($\gamma\_t = 1/\sqrt{t}$) is what enables the faster $O(t^{-1})$ convergence w.r.t. $\mathbb{E}[\text{Gap}^2]$. We will add a more explicit discussion of the two-phase behavior and the transition at $t\_0 = (LD/\sigma)^{4/3}$.
>
> ### On $\mathbb{E}[\text{Gap}^2]$ vs. $\mathbb{E}[\text{Gap}]^2$
>
> As noted in the introduction (lines 082-091, right column), we focus on $\mathbb{E}[\text{Gap}(x\_t)^2]$ because there may be a gap between $\mathbb{E}[\text{Gap}(x\_t)^2]$ and $(\mathbb{E}[\text{Gap}(x\_t)])^2$. This distinction is suggested by the results of Fiegel et al. (2025) for uncoupled learning in matrix games, where different rates are observed depending on which quantity is measured.
>
> We will add a more explicit discussion, clarifying that:
> - $\mathbb{E}[\text{Gap}(x\_t)^2] \ge (\mathbb{E}[\text{Gap}(x\_t)])^2$ by Jensen's inequality, so upper-bounding $\mathbb{E}[\text{Gap}(x\_t)^2]$, which we consider in this paper, is strictly harder
> - The squared gap captures the second moment, which provides stronger concentration guarantees
> - It remains open whether $\mathbb{E}[\text{Gap}(x\_t)]$ can converge faster than $\sqrt{\mathbb{E}[\text{Gap}(x\_t)^2]}$
>
> ### On "Regularized Monotone VI" Terminology
>
> The regularized VI defined by $V\_t(x) = V(x) + \omega\_t x$ is strongly monotone (not merely monotone), since $V$ is monotone and $\omega\_t x$ adds strong monotonicity. We will correct the terminology throughout.
>
> ### On the Existence of $U$ such that $\\|V(x)\\| \le U$
>
> Since $\mathcal{X}$ is compact and $V$ is continuous ($L$-Lipschitz), the bound $U = \max\_{x \in \mathcal{X}} \\|V(x)\\|$ is well-defined and finite by the extreme value theorem. We will add a note in the setup section.
>
> ### On the Phrasing "$\lambda \le L$ w.l.o.g."
>
> Thank you for pointing this out. The "w.l.o.g." phrasing is indeed misleading. As the reviewer notes, $\lambda \le L$ follows from $L$-Lipschitz continuity and $\lambda$-strong monotonicity: for any $x \neq y$,
> $$\lambda \\|x - y\\|^2 \le \langle V(x) - V(y), x - y \rangle \le \\|V(x) - V(y)\\| \cdot \\|x - y\\| \le L \\|x - y\\|^2,$$
> so $\lambda \le L$ is a consequence of the stated assumptions. We will fix the wording in the revised version.
>
> ### On the Density of the ROG Section
>
> We acknowledge that the ROG section is denser than the RG section. In the revised version, we will:
> - Expand the proof sketch in Section 6 to provide more intuition
> - Move some intermediate steps to the appendix
> - Add a comparison paragraph summarizing how the ROG analysis builds on the RG analysis
>
> ### On Additional Comments
>
> We will address the following in the revised version:
> - We will clarify the meaning of "fundamental gap between $\mathbb{E}[\text{Gap}^2]$ and $\mathbb{E}[\text{Gap}]^2$."
> - We will specify that $V\_t$ is strongly monotone when $V$ is monotone, and that strong monotonicity of $V$ makes regularization unnecessary ($\gamma\_t = 0$).
> - We will clarify that players can commit to their action rule at time $T+1$ based on information up to time $T$.
>
> We are grateful for these suggestions, which will improve the clarity of the paper.

---

> > ### Author Rebuttal · Reviewer_KeEa · 2026-04-03
> >
> > The authors have responded substantively to most of my comments, especially regarding the interpretation of the monotone-case rate, the distinction between last-iterate and average-iterate guarantees, the role of the squared gap metric, and several terminology/assumption clarifications. I appreciate the planned revisions to improve exposition. My main remaining concern is the part of the ROG analysis. The current reply promises added intuition, but I would still encourage the authors to make the key intermediate estimates more explicit.

---

> > > ### Author Response · Authors · 2026-04-04
> > >
> > > Thank you for the follow-up. We agree that adding intuition alone is not sufficient, and will make the key intermediate estimates in the ROG analysis more explicit in the main text. Specifically, in the proof sketch of Lemma 6.2, we plan to include:
> > >
> > > 1. The two projected gradient descent inequalities for $y\_{t+1}$ and $x\_t$ with their respective comparators, and how they are combined.
> > > 2. How the gradient difference term $\langle \hat{g}\_t - \hat{g}\_{t-1}, x\_t - y\_{t+1} \rangle$ is controlled via the bound
> > > $$\mathbb{E}[\\|\hat{g}\_{t-1} - \hat{g}\_t\\|^2] \le 6\sigma^2 + 6L^2(\mathbb{E}[\\|x\_{t-1} - y\_t\\|^2 + \\|x\_t - y\_t\\|^2]),$$
> > > and how this leads to the $12\sigma^2\eta\_t^2$ noise term in the recursion.
> > > 3. The contraction step showing how monotonicity yields the term $-\gamma\_t\\|y\_t - x\_t^*\\|^2$.
> > >
> > > We will ensure these steps are presented as displayed equations rather than inline, so that the logical flow will be easier to follow.
> > >
> > > We are again grateful for the many constructive comments and suggestions, which we believe will substantially improve the manuscript. If you have any additional questions or concerns, we would be happy to discuss them.

---

### Official Review · Reviewer_ANda · 2026-03-12

**Soundness:** 4
**Presentation:** 4
**Significance:** 3
**Originality:** 3
**Overall Recommendation:** 5
**Confidence:** 2

**Summary:**

Variational inequalities have many applications in game theory and AI. To solve them, regularized gradient methods use a monotone operator (V) that iteratively drives a sequence of points toward the solution. The gap function measures the distance to optimality: when it is zero, every feasible direction yields a zero inner product with (V), which characterizes optimality.
Exploiting the monotonicity of the operator, the analysis introduces an equivalent sequence of strongly monotone operators. The update rule can then be interpreted as gradient descent applied to a time-varying operator, generating a sequence of iterates that converges to the solution of the original problem.
Convergence bounds for this sequence are derived, showing that they depend on the step size and the regularization parameters, which can be chosen appropriately. With a suitable choice of parameters, the analysis also yields new results for regularized gradient and regularized optimistic gradient methods, both for fixed horizons and for strongly monotone operators.

**Compliance With Llm Reviewing Policy:**

Affirmed.

**Key Questions For Authors:**

-

**Limitations:**

yes

**Strengths And Weaknesses:**

The paper is well written. As a result, the subject, although abstract, is clearly presented, and the results are easy to follow. The proofs are complete and clearly developed.

---

> ### Author Rebuttal · Authors · 2026-03-30
>
> We thank Reviewer ANda for reading our paper carefully and for the supportive comments.
>
> We will incorporate improvements based on the feedback from other reviewers in the revised version, including clearer discussion of the connection between RG and ROG, a streamlined introduction, and improved presentation of Table 1. We are happy to address any additional questions.

---

> > ### Author Rebuttal · Reviewer_ANda · 2026-04-02
> >
> > I keep my current evaluation.

---

> > > ### Author Response · Authors · 2026-04-04
> > >
> > > Thank you for confirming the rebuttal. We again appreciate the time you spent reviewing our paper.

---

### Official Review · Reviewer_XgGx · 2026-03-12

**Soundness:** 2
**Presentation:** 2
**Significance:** 2
**Originality:** 2
**Overall Recommendation:** 3
**Confidence:** 3

**Summary:**

This paper studies last-iterate convergence for stochastic smooth monotone variational inequalities (VIs). It analyzes two single-call regularized methods: the regularized gradient (RG) method and the regularized optimistic gradient (ROG) method. The results give anytime last iterate guarantees without knowing the horizon and show that optimism improves convergence in the low-noise regime. They also provide synthetic experiments to verify the theory.

**Compliance With Llm Reviewing Policy:**

Affirmed.

**Final Justification:**

Due to the current disorganized presentation of this paper, I decide to maintain my current evaluation. I hope that the authors can improve the paper's clarity.

**Key Questions For Authors:**

Key questions:
1. Is there any lower bound with respect to the anytime convergence rate?

Minor comments:
1. The $\gamma_t$ and $\eta_t$ columns in Table 1 are not essential for a first reading and may cause confusion. Consider removing them from the main table or moving them to a separate table alongside the algorithm descriptions.
2. The RG and ROG updates are presented without derivation. A brief motivation would be better

**Limitations:**

Yes

**Strengths And Weaknesses:**

Strengths:
-  This paper appears technically solid.
- This paper is well-written and structured clearly.
-  This paper proves anytime last-iterate convergence for stochastic monotone VIs.

---

Weakness:
- There is limited discussion on the connection between RG and ROG.
- The introduction part may be lengthy, especially the contributions part.
-  Section 7 feels disconnected from the rest of the paper. Maybe it can be reorganized as an application section.

---

> ### Author Rebuttal · Authors · 2026-03-30
>
> We thank Reviewer XgGx for the constructive feedback. We hope the responses below adequately address the concerns.
>
> ### On the Connection Between RG and ROG
>
> We agree that the connection could be made more explicit. Both RG and ROG are built on the same regularized operator framework (Section 4). They use time-varying regularization $V\_t(x) = V(x) + \omega\_t x$ with $\omega\_t = \gamma\_t / \eta\_t$, and their analysis shares Lemmas 4.1 (stability) and 4.2 (gap decomposition).
>
> The key difference is that ROG adds optimistic gradient reuse on top of RG's regularization. In the ROG update, the gradient $\hat{g}\_t$ is used in both the $y\_{t+1}$ and $x\_{t+1}$ updates (Eq. (2)). This creates a structure where consecutive gradient errors partially cancel, leading to a tighter recursion. Concretely:
> - The RG recursion (Lemma 5.2) has a noise term $2G^2\eta\_t^2$, where $G^2$ bounds $\mathbb{E}[\\|\hat{g}\_t\\|^2]$ and includes both $\\|V(x\_t)\\|^2$ and $\sigma^2$. Even when $\sigma \to 0$, the $\\|V(x\_t)\\|^2$ contribution remains.
> - The ROG recursion (Lemma 6.2) has a noise term $12\sigma^2\eta\_t^2$, which depends only on the variance $\sigma^2$ and vanishes when $\sigma \to 0$.
>
> This structural difference enables ROG's variance-adaptive rate: when $\sigma$ is small, the noise term becomes negligible and ROG achieves $O(t^{-1})$, whereas RG remains at $O(t^{-2/5})$ regardless of $\sigma$.
>
> We will add a dedicated paragraph in the revised version comparing the two algorithms and their recursion structures.
>
> ### On the Introduction Length
>
> We acknowledge that the introduction is lengthy. The current structure was chosen to make the paper accessible to readers from both the optimization and game theory communities. We agree that some material could be reorganized. In the revised version, we plan to:
> - Move part of the related work discussion to a separate section or to the appendix
> - Streamline the motivational discussion
>
> ### On Section 7 (Regret Analysis)
>
> We understand your concern. One motivation for including Section 7 is the question of whether no-regret learning and last-iterate convergence can be achieved simultaneously, which was studied by Cai and Zheng (2023, "Doubly optimal no-regret learning in monotone games") in the noiseless VI setting. Section 7 provides some insight into this question in the stochastic setting: our RG method achieves both sublinear regret ($O(T^{4/5})$) and anytime last-iterate convergence, though the regret rate is worse than the standard $O(\sqrt{T})$. This suggests room for future improvement in reconciling these two objectives, and we believe this perspective gives Section 7 additional significance beyond a standalone regret bound.
>
> We acknowledge that this motivation is not clearly communicated in the current manuscript. In the revised version, we will add a discussion connecting Section 7 to the above context, or move the regret analysis to the appendix if space allows for a cleaner main body.
>
> ### On Table 1: Separating $\gamma\_t$ and $\eta\_t$ Columns
>
> We agree that including $\gamma\_t$ and $\eta\_t$ in Table 1 may cause confusion on a first reading. We will consider moving the parameter columns to a separate table in the algorithm sections and keeping Table 1 focused on convergence rates only. Thank you for this suggestion.
>
> ### On Lower Bounds for the Anytime Convergence Rate
>
> We do not currently have matching lower bounds, and we acknowledge this as an important open problem (discussed in Section 8). In fact, a lower bound for bandit feedback, which is a related setting, has only been established very recently by Fiegel et al. (2025).
>
> Note that, as discussed in Remark D.1 in the appendix, the stochastic gradient feedback considered in our work is more informative than bandit feedback, and therefore their lower bound does not apply to our setting. Specifically, under bandit feedback, the KL divergence between feedback distributions for two similar games scales as $(\delta\_t \cdot \epsilon)^2$, where $\delta\_t$ is the approximation quality. Under stochastic gradient feedback, the KL divergence becomes $O(\epsilon^2)$ independent of $\delta\_t$, which breaks the lower bound argument. Establishing lower bounds for the stochastic gradient feedback setting appears to require new techniques. We believe this is an important direction for future work.
>
> ### On Simplifying the RG and ROG Derivations
>
> We will add clearer algorithmic motivation for both methods in the revised version:
> - For RG: we will make explicit that the update (1) is equivalent to projected gradient descent on the regularized operator $V\_t$
> - For ROG: we will explain that it combines RG's regularization with optimistic gradient technique, where the gradient $\hat{g}\_t$ is reused to anticipate the next gradient direction

---

> > ### Author Rebuttal · Reviewer_XgGx · 2026-04-04
> >
> > I thank the authors for the detailed responses. However, I decide to keep my current evaluation. I believe that the authors will further improve the manuscript.

---

> > > ### Author Response · Authors · 2026-04-04
> > >
> > > Thank you for confirming that the concerns have been resolved and for the encouragement. We will make sure to further improve the manuscript in the revised version, incorporating the suggestions discussed above. We appreciate the time and effort you spent on reviewing our paper.

---

### Official Review · Reviewer_SNgF · 2026-03-15

**Soundness:** 4
**Presentation:** 3
**Significance:** 2
**Originality:** 2
**Overall Recommendation:** 4
**Confidence:** 4

**Summary:**

The paper studies stochastic monotone variational inequalities through the lens of single-call regularized methods, with particular emphasis on uncoupled dynamics that are natural in multi-agent settings. The authors analyze two algorithms, the regularized gradient (RG) method and the regularized optimistic gradient (ROG) method, both of which incorporate a time-varying regularization term that stabilizes the dynamics. A central feature of the paper is that the convergence guarantees are given for the last iterate, which is the most meaningful notion in many applications and is generally harder to establish than averaged convergence.

The main technical results provide bounds on the squared gap function in both the anytime and fixed-horizon settings. For RG, the paper establishes an anytime last-iterate rate of order O(t^{-2/5}), as well as an improved fixed-horizon rate when the horizon is known in advance. For ROG, the main result is a variance-adaptive anytime guarantee of order O(\sigma^{4/5}t^{-2/5}+t^{-1}), showing that the method interpolates between noisy and nearly noiseless regimes. The paper also gives guarantees in the strongly monotone setting, where the rates improve further.

Conceptually, the analysis is based on introducing a regularized operator V_t(x)=V(x)+\omega_t x, which turns the original monotone problem into a sequence of strongly monotone surrogate problems. The authors then track the distance of the iterates from the solution of these surrogate problems and derive convergence through a bias-variance tradeoff induced by the regularization.

**Compliance With Llm Reviewing Policy:**

Affirmed.

**Final Justification:**

I will give my weak positive vote. I believe that it needs better argumentation about the necessities of any-time guarantee.

**Key Questions For Authors:**

At a technical level, I suspect this is the most important point for evaluating the contribution: what exactly is the new ingredient beyond the classical recipe of defining a suitable decaying potential and controlling its evolution across iterations? As currently written, I can see the framework, but I do not yet clearly see the decisive step that makes the analysis qualitatively different from prior monotone VI or optimistic-gradient arguments. Making this distinction more explicit would help the reader understand whether the contribution is primarily a clean consolidation of known ideas, or whether there is a truly new proof insight underlying the anytime guarantees.

In more details, I think deserves a more explicit discussion is the relationship between the present analysis and classical potential/Lyapunov-based convergence proofs. My impression is that this technical distinction is central for appreciating the contribution of the paper. At the moment, I do not fully see what the crucial proof innovation is that goes beyond the standard paradigm of choosing a decaying potential and balancing stability, bias, and noise terms. Clarifying this point would substantially strengthen the paper, especially for readers trying to understand the real source of novelty.

**Limitations:**

Well explained

**Strengths And Weaknesses:**

Overall, the paper is clearly written, well organized, and the results are presented in a clean and accessible way.
The clearest strength of the paper is its presentation: the work is very well written, the algorithmic framework is easy to follow, and the technical development is structured in a transparent way. In particular, the focus on last-iterate guarantees in the anytime setting is certainly meaningful. That said, from my perspective, the main limitation is that I did not feel the paper delivers a truly significant new technical insight. Especially even if one focuses on the anytime results, which seem to be the main contribution, the progress feels somewhat incremental rather than conceptually transformative. I also believe that, in the strongly monotone VI setting, one would ideally hope for an exponential rate, not to arbitrary small polynomial while $\sigma\to 0$ so the guarantees there felt somewhat less compelling than they could have been.

---

> ### Author Rebuttal · Authors · 2026-03-30
>
> We thank Reviewer SNgF for the careful review. We hope the following clarifies our contributions and addresses the concerns raised.
>
> ### On the Significance of This Work
>
> Our primary goal is not solely to introduce new proof techniques, but to establish what can be achieved with simple, natural algorithms for an important problem where the existing literature is limited. The closest prior results by Abe et al. (2024, 2025) employ complex algorithmic designs and achieve fixed-horizon rates of $O(T^{-1/10})$ and $O(T^{-1/7})$, respectively. Our work shows that simpler single-call regularized gradient methods achieve $O(t^{-1/5})$ in the stronger anytime setting, without requiring knowledge of $T$.
>
> In retrospect, the manuscript does not communicate this motivation clearly enough. We will rewrite the relevant parts of the introduction in the revised version. We also understand the importance of technical novelty at ICML and will make the technical contributions more visible. We elaborate below.
>
> ### On Technical Novelty: RG Analysis (Non-Strongly Monotone, Anytime)
>
> The high-level analysis framework shares structural similarities with prior work, notably Appendix C of Cai et al. (2023, NeurIPS: "Uncoupled and convergent learning in two-player zero-sum Markov games with bandit feedback"). Nevertheless, the new techniques introduced in our analysis lead to an essential improvement in the convergence rate: the $O(t^{-1/3})$ rate achievable by the existing analysis and parameter choices is improved to $O(t^{-2/5})$.
>
> The improvement comes from a refinement in the proof of Lemma 3.1 (the RG recursion). In bounding $\\| x\_t^* - x\_t \\|^2$, we expand it into $\\| x\_t^* - x\_{t-1}^* \\|^2 + \\| x\_{t-1}^* - x\_t \\|^2$ plus a cross term, and apply Cauchy--Schwarz followed by AM-GM with a parameter tuned to $\gamma\_t$. This yields a tighter contraction coefficient of $(1 - \gamma\_t)$ on $a\_t$. The analogous step in prior work bounds the same quantity more loosely as $O(\\| x\_t^* - x\_{t-1}^* \\|)$, leading to the weaker $O(t^{-1/3})$ rate. We refer to the proof of Lemma 3.1 (Appendix B) for the details.
>
> This refinement was not identified in the prior literature to the best of our knowledge, and is potentially applicable to other algorithms in the same framework, including those in Cai et al. (2023) and Zhao et al. (ICML 2025: "Learning Imperfect Information Extensive-form Games with Last-iterate Convergence under Bandit Feedback"). Extending the technique to KL or Bregman divergence (needed for these settings) is nontrivial, but the potential scope is broader than our specific setting. Note also that Cai et al. (2023) study two-player zero-sum (Markov) games with bandit feedback, whereas our work considers monotone VIs under stochastic gradient feedback, so the two works are complementary.
>
> ### On Technical Novelty: ROG Analysis
>
> The ROG result (Theorem 4.1) is, to our knowledge, the first variance-adaptive last-iterate convergence guarantee for stochastic monotone VIs. The rate $O(\sigma^{4/5} t^{-2/5} + t^{-1})$ interpolates between the high-noise and noiseless regimes via optimistic gradient reuse. A key challenge in the analysis is the choice of potential. The natural candidate $\mathbb{E}[ \\| x\_t - x\_t^* \\|^2]$ from the RG analysis does not lead to a closed recursion for ROG due to the two-sequence structure. The potential we use, $a\_t = \mathbb{E}[ \\| y\_t - x\_{t-1}^* \\|^2 + \frac{1}{2} \\| x\_{t-1} - y\_t \\|^2]$, was found through trial and error to yield a recursion (Lemma 4.1) compatible with the optimistic structure under time-varying regularization. This choice additionally guarantees the convergence of both $x\_t$ and $y\_t$, as noted in the paper. The analysis also requires a two-regime parameter design that transitions at $t\_0 = (LD/\sigma)^{4/3}$, and ensuring the convergence bound holds across both regimes through a unified induction is technically involved. We will clarify these points in the revised version.
>
> ### On the Strongly Monotone Case and Exponential Rates
>
> We agree that in the noiseless case ($\sigma = 0$), an exponential rate for the initial condition term should be achievable. Our current ROG result gives $O(\sigma^2/(\lambda^2 t) + t^{-c})$ for any $c \ge 2$, where $t^{-c}$ is not exponential. This is because our result prioritizes anytime guarantees that hold uniformly for all $\sigma \ge 0$, which constrains the parameter design. A natural approach would be to switch parameter schedules between a low-variance regime (targeting exponential decay) and a high-variance regime, but reconciling the two frameworks at the transition point is difficult. The polynomial rate $t^{-c}$ with arbitrarily large $c$ is the best we achieve under a unified schedule. We consider this an interesting open problem and will discuss it in the revised version.

---

> > ### Author Rebuttal · Reviewer_SNgF · 2026-04-02
> >
> > I will keep my evaluation

---

> > > ### Author Response · Authors · 2026-04-04
> > >
> > > Thank you for confirming our rebuttal and for noting that your concerns have been addressed. We sincerely appreciate the time you dedicated to reviewing our paper and the many constructive comments provided through your careful review. We are confident that the manuscript will be improved thanks to these comments. If you have any further questions or concerns, we would be happy to discuss them.

---

### Decision · Program_Chairs · 2026-04-30

**Decision:**

Accept (regular)

**Comment:**

The paper establishes new last-iterate convergence rates for the regularized gradient (RG) and regularized optimistic gradient (ROG) methods in the setting of stochastic variational inequalities over a bounded domain. The analysis assumes a Lipschitz (strongly) monotone operator and access to an unbiased stochastic oracle with bounded variance. Additionally, the operator itself is assumed to be bounded, which is a mild condition given the boundedness of the domain. Notably, the authors derive both fixed-horizon and horizon-free (anytime) convergence rates.

The reviewers generally agreed on the soundness of the results, the clarity and organization of the paper (with the exception of Reviewer XgGx), and the novelty of the obtained rates. Some disagreement remained regarding the degree of technical novelty and the practical significance of the anytime guarantees. The authors addressed these concerns carefully and substantively in their rebuttal.

Taking into account the reviews, the authors’ responses, the discussion among reviewers, and my own assessment, I find that the paper makes a meaningful theoretical contribution. Improving last-iterate convergence rates is an important and active line of research, and the results presented here appear to advance the state of the art. The authors have also been appropriately transparent about the limitations of their work, both in the manuscript and in the rebuttal. Provided that these clarifications are clearly reflected in the final version, I recommend acceptance.